# Sir2 and Fun30 regulate ribosomal DNA replication timing via MCM helicase positioning and nucleosome occupancy

**Carmina Lichauco[1†], Eric J Foss[1†], Tonibelle Gatbonton-Schwager[1], Nelson F Athow[1], Brandon Lofts[1], Robin Acob[1], Erin Taylor[1], James J Marquez[1], Uyen Lao[1], Shawna Miles[1], Antonio Bedalov[1,2]***

[1]Translational Science and Therapeutics Division, Human Biology Division, Fred Hutchinson Cancer Center, Seattle, United States; [2]Department of Biochemistry and Department of Medicine, University of Washington, Seattle, United States

*For correspondence:
abedalov@fredhutch.org

†These authors contributed equally to this work

**Competing interest:** The authors declare that no competing interests exist.

## eLife Assessment

This **valuable** study is a detailed investigation of how chromatin structure influences replication origin function in yeast ribosomal DNA, with a focus on the role of the histone deacetylase Sir2 and the chromatin remodeler Fun30. The paper shows that Sir2 does not affect origin licensing but rather affects local transcription and nucleosome positioning which correlates with increased origin firing. Overall, the evidence is **convincing** and the model is plausible.

## Abstract

The association between late replication timing and low transcription rates in eukaryotic heterochromatin is well known, yet the specific mechanisms underlying this link remain uncertain. In *Saccharomyces cerevisiae*, the histone deacetylase Sir2 is required for both transcriptional silencing and late replication at the repetitive ribosomal DNA (rDNA) arrays. We have previously reported that in the absence of *SIR2*, a de-repressed RNA PolII repositions MCM replicative helicases from their loading site at the ribosomal origin, where they abut well-positioned, high-occupancy nucleosomes, to an adjacent region with lower nucleosome occupancy. By developing a method that can distinguish activation of closely spaced MCM complexes, here we show that the displaced MCMs at rDNA origins have increased firing propensity compared to the nondisplaced MCMs. Furthermore, we found that both activation of the repositioned MCMs and low occupancy of the adjacent nucleosomes critically depend on the chromatin remodeling activity of *FUN30*. Our study elucidates the mechanism by which Sir2 delays replication timing, and it demonstrates, for the first time, that activation of a specific replication origin in vivo relies on the nucleosome context shaped by a single chromatin remodeler.

## Introduction

*SIR2* is the founding member of a family of NAD-dependent protein deacylases called sirtuins, which have important effects on transcription, replication, chromatin structure, and metabolism in organisms from yeast to humans (*Houtkooper et al., 2012*; *Shahgaldi and Kahmini, 2021*). *SIR2* was initially identified in budding yeast in 1979 on the basis of its role in transcriptional repression at the silent mating type loci (*Haber and George, 1979*; *Klar et al., 1979*; *Rine et al., 1979*), and it was therefore dubbed a 'Silent Information Regulator'. The silent mating type loci are heterochromatic regions that contain nontranscribed copies of the small open reading frames that determine yeast mating type, and deletion of *SIR2* not only activates their transcription but also abolishes their heterochromatic

structure (*Ivy et al., 1986*; *Rine and Herskowitz, 1987*; *Weiss and Simpson, 1998*; *Ravindra et al., 1999*; *Haber, 2012*). It was later discovered that, in addition to regulating transcription at the silent mating type loci, Sir2 also regulates DNA replication, and it does so not only at the silent mating type loci but also at the two other heterochromatic regions of the yeast genome, namely at telomeres and the rDNA (*Pasero et al., 2002*; *Yoshida et al., 2014*). It is this last activity of Sir2, namely its effect on replication at the rDNA, that is the focus of this report.

The rDNA, which encodes the structural RNA components of the ribosome, is composed of approximately 150 tandemly arranged 9.1 kb repeats (*Woolford and Baserga, 2013*). At almost 1.4 megabases, the rDNA is too large to be replicated passively, and thus it must contain its own origins of replication. This is achieved by the organization shown in *Figure 1A*: The ribosomal RNAs are produced as two major species, known as the 35S and 5S, which are transcribed by RNA PolI and RNA PolIII, respectively (*Fernández-Pevida et al., 2015*). Between the 5' ends of these genes is an origin of replication, and immediately to the left of this origin is a short Sir2-repressed RNA PolII-transcribed noncoding transcript called C-pro (*Kobayashi and Ganley, 2005*; *Li et al., 2006*; *Vasiljeva et al., 2008*), which we will discuss further below. In wild-type (WT) cells, rDNA origins fire in the second half of S phase, making the rDNA among the later replicating regions of the genome, while in *sir2* mutants, they fire early in S phase (*Pasero et al., 2002*; *Yoshida et al., 2014*; *Foss et al., 2017*).

An indication of the biological significance of maintaining the late replication status of the rDNA came from a genome-wide association study aimed at identifying naturally occurring genetic variants that affect replicative lifespan (*Kwan et al., 2013*). Replicative lifespan in budding yeast is defined as the number of daughter cells that a mother cell is able to produce and is therefore a measure of a cell's replicative capacity at birth. By measuring lifespan in segregants from a cross between two genetically diverse strains of yeast, Kwan et al. were able to identify a point mutation at the rARS that (1) decreased origin activity at the rDNA; (2) increased origin activity elsewhere in the genome; and (3) prolonged replicative lifespan. They interpreted these results as reflecting competition for replication resources between the origins in two distinct compartments of the nucleus, namely the heterochromatic repetitive nucleolus or rDNA and the nonrepetitive sequences that make up the bulk of the yeast genome. This competition is exacerbated by the perturbation of nucleolar heterochromatin caused by the absence of *SIR2* (*Yoshida et al., 2014*; *Foss et al., 2017*). Most replication outside of the rDNA initiates from approximately 300 replication origins, and thus the 150 origins at the rDNA make up approximately one-third of total origins. Increased replication resources directed toward the rDNA might therefore be expected to create a significant strain on replication in the rest of the genome.

Replication origins are known to compete for limiting resources at the level of origin firing. Broadly speaking, replication origin activity is regulated at two levels, namely origin licensing and origin firing (*Remus and Diffley, 2009*). Origin licensing refers to loading of the replicative helicase in G1, and only origins that have been licensed are capable of being used in the subsequent S phase. Licensing occurs when an origin recognition complex, Orc1-6, that is bound to an 11–17 base pair so-called ARS consensus sequence (ACS) within the origin loads two hexameric helicases, each consisting of a ring of subunits, MCM2-7, that encircles the DNA (*Bell and Stillman, 1992*; *Francis et al., 2009*; *Bell and Botchan, 2013*; *Deegan and Diffley, 2016*; *Deegan et al., 2016*; *Douglas et al., 2018*; *Lewis et al., 2022*). The mechanism by which Orc recognizes origins and loads an MCM double-hexamer has been elucidated in great detail (*Bell and Labib, 2016*; *Costa and Diffley, 2022*; *Hu and Stillman, 2023*). DNA replication initiates from only a subset of these licensed origins, namely those where the helicase complex is activated, a process referred to as origin firing (*Rhind and Gilbert, 2013*). Origin firing is triggered by activation of two kinase signaling pathways, DDK and CDK, which then cause each hexameric MCM helicase to encircle a single strand of DNA and track in opposite directions in front of the replisome, thereby establishing bidirectional replication (*Heller et al., 2011*; *Duncker, 2016*; *Li et al., 2023*). Firing factors are not present in sufficient quantities to activate all licensed origins simultaneously, and thus origins are activated in waves, with firing factors being recycled until the entire genome has been replicated (*Mantiero et al., 2011*; *Tanaka et al., 2011*). Determining the criteria by which origins are prioritized for activation remains an area of active research.

Importantly, if the competition between the origins at the rDNA and those in the rest of the genome is tilted in favor of the rDNA, either by a point mutation in the rARS or by deletion of *SIR2*, replication gaps between distantly spaced origins outside of the rDNA persist late into S phase, and genome

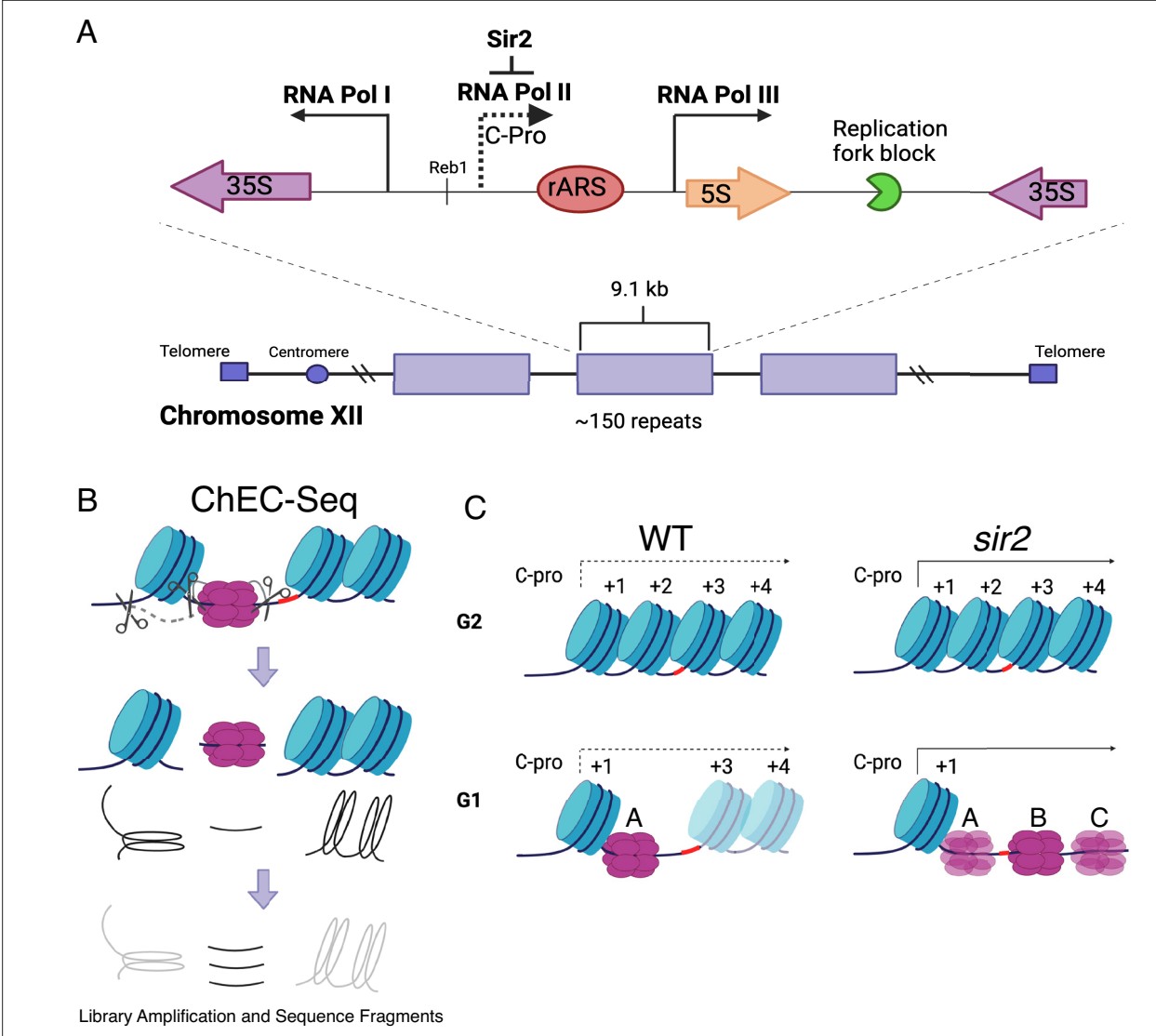

**Figure 1.** Ribosomal DNA (rDNA) structure, chromatin endogenous cleavage (ChEC), and MCM displacement in *sir2*. (**A**) The 1.4 megabase rDNA region on chromosome XII is composed of approximately one hundred and fifty 9.1 kb repeats. Each repeat encodes both the 35S and 5S ribosomal RNAs (rRNA), which are transcribed by RNA polymerase I (PolI) and RNA polymerase III (PolIII), respectively. The ribosomal origin of replication (rARS) is located between the 5' ends of these genes. The C-pro transcript, which is suppressed by Sir2, initiates approximately 200 base pairs from the rARS. The unidirectional replication fork block, which functions to prevent collision between replication and transcription machinery, is depicted in green. The probe used in Southern blots in *Figure 3* to assess licensing is marked by *. (**B**) In ChEC-seq, micrococcal nuclease (MNase; depicted as scissors) is fused to the protein of interest, in this case MCM2. The double-hexameric MCM helicase complex, MCM2-7, is depicted in purple, and nucleosomes are in blue. Cleavage is induced by addition of calcium to permeabilized cells. Due to the proximity of nucleosome entrance and exit sites, MCM2-ChEC can reveal not only the binding site of the MCM complex but also that of the adjacent nucleosome. (**C**) Depiction of nucleosomes and MCM complex in wild-type and *sir2* in G1 and G2. Nucleosomes are numbered with respect to C-pro transcription. De-repression of C-pro transcription in *sir2* causes RNA PolII to displace the MCM complex to the right. The three different MCM helicase complexes depicted in the bottom right panel are intended to convey the three most prominent locations for the complex in *sir2*; this is not intended to indicate that there is ever more than one MCM complex in the same rDNA repeat. Created with BioRender.com.

The online version of this article includes the following source data and figure supplement(s) for figure 1:

**Figure supplement 1.** Computer-generated visualization of nondisplaced and displaced MCM complexes, as determined by MCM2-chromatin endogenous cleavage (ChEC).

**Figure supplement 1—source data 1.** Source data for *Figure 1—figure supplement 1*.

duplication may remain incomplete when the cell enters mitosis (*Foss et al., 2017*). Indeed, the observation that weakening the rARS by a point mutation can suppress the short replicative lifespan characteristic of *sir2* mutants suggests that these replication gaps may be the ultimate cause of 'replicative death'. Teleologically, one can rationalize the prioritization of replication of the unique areas of the genome over the repetitive and heterochromatic areas, because (1) there are mechanisms available for repair in repetitive regions, like single-strand annealing, that are not available in unique regions, and they may therefore be able to survive lesions created when mitosis precedes the completion of replication; and (2) the cell is able to tolerate variation in copy number at the rDNA, but not deletion of protein-coding sequences in unique regions. Consistent with this view of the utility of assigning a low priority to replication of repetitive sequence, heterochromatic regions are also late replicating in mammalian cells, and disease states have been associated with disruption of this pattern (*Donley and Thayer, 2013*).

Despite its biological importance, the mechanism by which Sir2 suppresses replication initiation is poorly understood. Clues to Sir2's role in replication may be gleaned from its role in transcription, although here, too, our understanding is limited. Sir2's ability to repress transcription at the silent mating type loci has been postulated to follow from its role in chromatin compaction. Experiments demonstrating *SIR2*-dependent protection of heterochromatin from digestion with endonucleases or methylation by bacterial DNA methyltransferases support the notion that DNA in heterochromatin is generally less accessible (*Gottschling, 1992*; *Singh and Klar, 1992*; *Loo and Rine, 1994*; *Ansari and Gartenberg, 1999*). Thus, a widely publicized model attributes Sir2's ability to inhibit transcription to blocking access of RNA PolII to the DNA, although a simple model of indiscriminate steric occlusion cannot account for this phenomenon (*Chen and Widom, 2005*; *Lynch and Rusche, 2009*; *Kitada et al., 2012*; *Bondra and Rine, 2023*). Whether Sir2's effect on replication is similarly the result of chromatin compaction limiting access to enzymes involved in DNA metabolism is not known.

A study examining the distribution of MCM helicase complexes at the rDNA suggests that, rather than a global manifestation of lack of accessibility of the rDNA, the early activation of replication at the rDNA in the absence of Sir2 may reflect a more localized phenomenon (*Foss et al., 2019*). In this study, MCM binding was assessed at nucleotide-level resolution by fusing a subunit of MCM to micrococcal nuclease (MNase), a technique known as chromatin endogenous cleavage, or 'ChEC' (*Figure 1B*; *Schmid et al., 2004*; *Zentner et al., 2015*). Permeabilization of cells with detergent followed by the addition of calcium activates MNase, thereby generating approximately 65 base pair fragments of DNA under MCM double-hexamers that can be identified by high-throughput sequencing. This analysis revealed that, while deletion of *SIR2* did not elicit an obvious change in the overall levels of licensing at the rDNA, it led to a dramatic redistribution of MCM helicase complexes: In WT cells, the helicase complexes are located almost exclusively immediately adjacent to a high-occupancy well-positioned nucleosome approximately 130 base pairs to the left of the ACS. In contrast, the MCM complexes in *sir2* mutants are largely displaced to the right into an area with lower nucleosome occupancy (see *Figure 1C*, *Figure 1—figure supplement 1*). This displacement appears to be caused by the de-repression of C-pro transcription in the absence of *SIR2*, which results in RNA PolII pushing the MCM double-hexamer into the relatively nucleosome-free region. This observation is provocative in light of studies demonstrating that the arrangement of MCM-proximal nucleosomes can have a dramatic effect on MCM activation (*Azmi et al., 2017*). Specifically, by using chromatin remodeling enzymes (CREs) to create a variety of nucleosomal arrangements on an origin-containing fragment of DNA, Azmi et al. demonstrated that activation of an MCM double-hexamer loaded at that origin could be inhibited by nucleosomes arranged by specific CREs. In another study (*Chacin et al., 2023*), a mutant version of Orc1 that retains the ability to load MCM but alters phasing of MCM adjacent nucleosomes was shown to inhibit DNA replication in vivo and in vitro, consistent with a critical role for nucleosome organization at origins in the initiation of replication.

Taken together, the observations above led us to hypothesize that (1) relocation of MCM helicases from regions of high nucleosome occupancy to nearby regions of low nucleosome occupancy promotes early activation of replication origins in *sir2*; (2) this local difference in nucleosome occupancy is the result of modulation of nucleosomal architecture by one or more CREs; and therefore (3) screening CREs for the ability to suppress the early rDNA replication phenotype in *sir2* mutants might allow us to identify the relevant CRE. In this report, we show that the displaced MCM complexes fire before their nondisplaced counterpart and that increased nucleosome occupancy near the displaced

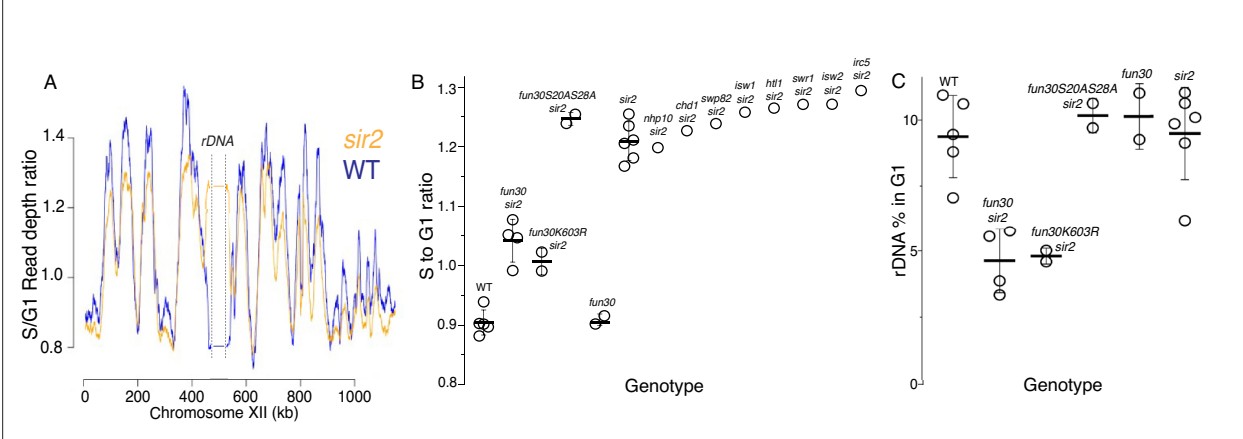

**Figure 2.** S-seq applied to ribosomal DNA (rDNA) replication timing and copy number determination. (**A**) S-seq replication profile of Chromosome XII. The region in the middle noted as the rDNA has been collapsed. Note that the rDNA replicates much earlier in *sir2* (16559) (orange) than in wild-type (16535) (blue). (**B**) rDNA replication timing in double mutants between *sir2* and various chromatin remodeling enzymes: *nhp10* (16729), *chd1* (16725), *swp82* (17061), *isw1* (16724), *htl1* (17059), *swr1* (16728), *isw2* (16673), and *irc5* (16723). *fun30-S20A S28A* (17113 and 17114) is non-phosphorylatable, and *fun30-K603R* (17345 and 17346) is catalytically inactive. The S to G1 ratio values (mean ± SD) were 0.90±0.02 for WT, 1.04±0.04 for *fun30sir2* (16909), 1.01±0.02 for *fun30K603R,* and 1.21±0.03 for *sir2* (p<0.001 for WT vs *sir2*, *sir2* vs *fun30 sir2*, and *sir2* vs *fun30K603Rsir2* using t-test). (**C**) Effect of *sir2* and *fun30* mutation on rDNA size, as determined from the fraction of G1 sequencing reads that arise from 450 to 470 kb on chrXII. The values (% mean ± SD) were 9.4±1.6 for WT, 4.6±1.2 for *fun30sir2*, 4.8±0.3 for *fun30K603R,* and 9.2±1.8 for *sir2* (p<0.001 for WT vs *sir2*, *sir2* vs *fun30sir2,* and *sir2* vs *fun30K603R sir2* using t-test).

The online version of this article includes the following source data and figure supplement(s) for figure 2:

**Source data 1.** Source data for plots displayed in *Figure 2A–C*.

**Figure supplement 1.** Ribosomal DNA (rDNA) size as determined by qPCR.

**Figure supplement 2.** Changes in ribosomal DNA (rDNA) size of *fob1* strains with continuous passaging.

**Figure supplement 3.** Ribosomal DNA (rDNA) replication timing in *fob1* strains.

**Figure supplement 1—source data 1.** Source data for the plot displayed in *Figure 2—figure supplement 1*.

**Figure supplement 2—source data 1.** Source data for the plot displayed in *Figure 2—figure supplement 2*.

**Figure supplement 3—source data 1.** Source data for the plot displayed in *Figure 2—figure supplement 3*.

MCMs in the absence of Fun30 CRE suppresses their early activation. This is the first in vivo demonstration of the modulation of an origin activity by a specific chromatin remodeler in vivo, and has implications for the broader relationship between transcription and replication timing.

## Results

### *Fun30* chromatin remodeling activity is required for early rDNA replication in *sir2* mutant

To identify CREs required for the early replication of the rDNA in *sir2* mutants, we needed a method to determine replication timing, and for this we turned to S-seq. In this widely used technique, G1 and S phase populations of cells from a log phase culture are sorted using flow cytometry and their DNA subjected to genomic sequencing (*Müller et al., 2014*; *Batrakou et al., 2020*). Replication timing can then be inferred from read depths in the S phase fraction, because regions that replicate early in S phase will be present in two copies in most cells, while regions that replicate late will be present mostly as a single copy. S phase read depths are normalized to those in G1 to account for differences in DNA copy numbers, which is particularly important for repetitive loci whose copy number can vary, such as rDNA. As shown in *Figure 2*, which depicts the S-seq profile of chromosome 12, relative replication timing (Trel) values using this approach typically range from 1.4 to 0.8 for early- and late-replicating regions, respectively, with a value of 1 indicating that a locus replicates at the same time as the genome-wide average (see Materials and methods for specific calculation). The late replication of the rDNA, defined as coordinates 450,000–470,000 on chrXII, in WT is reflected in a Trel value of 0.89,

whereas this ratio increases to 1.2 upon deletion of *SIR2* (*Figure 2A and B*). Using this method, we showed that a point mutation in the ACS of the rARS suppresses early replication of the rDNA in *sir2*, while promoting replication in the rest of the genome (*Kwan et al., 2013*; *Foss et al., 2017*). To identify potential chromatin remodelers required for early rDNA replication in *sir2*, we measured rDNA replication timing in double mutants in *sir2* and each of nine nonessential CRE subunits (Chd1, Fun30, Htl1, Irc5, Isw1, Isw2, Nhp10, Swr1, and Swp82). Deletion of *FUN30*, but no other gene, delayed replication of the rDNA in a *sir2* background, shifting the Trel value from 1.2 to 1.06, while having had no significant effect in *SIR2* cells (0.89 for WT and 0.91 for *fun30*) (*Figure 2B*).

Because Fun30 can form a complex with the replication scaffold protein Dpb11, it was possible that Fun30 played a structural rather than an enzymatic role in regulating rDNA replication. However, we found that a point mutation that abolishes Fun30 ATPase activity, K603R (*Neves-Costa et al., 2009*; *Eapen et al., 2012*), also suppresses early rDNA replication (*Figure 2B*), arguing that Fun30 instead promotes rDNA replication through its enzymatic activity in nucleosome remodeling. Fun30 is also known to play a role in the DNA damage response; specifically, phosphorylation of Fun30 on S20 and S28 by CDK1 targets Fun30 to sites of DNA breaks, where it promotes DNA resection (*Chen et al., 2016*; *Bantele et al., 2017*). To determine whether the replication phenotype that we observed might be a consequence of Fun30's role in the DNA damage response, we tested non-phosphorylatable mutants for the ability to suppress early replication of the rDNA in *sir2*; these mutations had no effect on the replication phenotype (*Figure 2B*), arguing against a primary role for Fun30 in DNA damage repair that somehow manifests itself in replication.

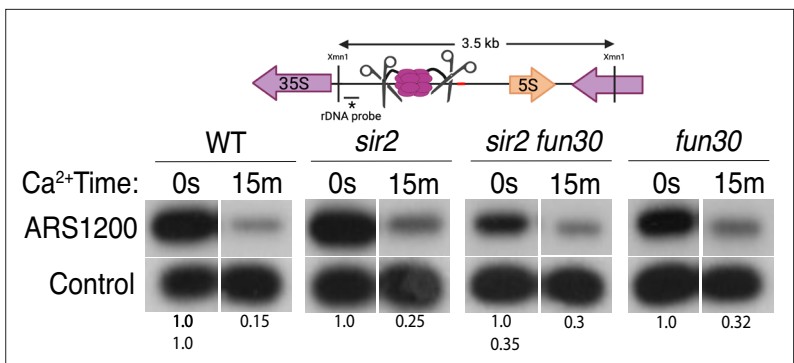

**Figure 3.** Licensing at the rARS, as determined by Southern blot. Activation of MCM-micrococcal nuclease (MNase) in G1-arrested wild-type (WT) (16747), *sir2* (16769), *sir2fun30* (17257), and *fun30* (17256) cells at the rARS with calcium will eliminate the 3.5 kb XmnI fragment (upper panel). *PIK1*, a single-copy gene in which we detect no MCM binding, is used as a loading control. Normalized ARS1200 band intensity at 15 m is expressed relative to time 0. Quantitation of the uncut band was used to infer relative ribosomal DNA (rDNA) array size in *sir2 fun30* mutant at 0.35 relative to WT.

The online version of this article includes the following source data and figure supplement(s) for figure 3:

**Source data 1.** PDF file containing original Southern blots displayed in *Figure 3* indicating the relevant bands, treatments and band sizes.

**Source data 2.** Original files for Southern blot analysis displayed in *Figure 3*.

**Source data 3.** Quantification of bands in Southern blots and calculations of MCM loading displayed in *Figure 3*.

**Figure supplement 1.** Licensing at the rARS using Southern blot (replica of *Figure 3*).

**Figure supplement 1—source data 1.** PDF file containing original Southern blots displayed in *Figure 3—figure supplement 1* indicating the relevant bands and treatments.

**Figure supplement 1—source data 2.** Original files for Southern blot analysis displayed in *Figure 3—figure supplement 1*.

**Figure supplement 1—source data 3.** Quantification of bands in Southern blots and calculations of MCM loading (*Figure 3—figure supplement 1*).

## *Fun30* chromatin remodeling activity is required for maintaining rDNA array size in *sir2* mutant

We next wanted to determine whether *FUN30* affects rDNA array size. The repetitive nature of the rDNA allows it to fluctuate in size, and mutations that impair DNA replication, either globally or in a manner specific to the rDNA, are associated with shrinkage of the rDNA. For example, the rDNA in a WT strain in the S288c background consists of 150 copies, but a point mutation at the ACS that impairs Orc binding causes the rDNA to shrink to just 80 copies (*Kwan et al., 2013*). rDNA shrinkage in response to global replication defects is thought to reflect the fact that the rDNA, which typically comprises approximately 10% of genomic DNA, imposes a significant replicative burden on the cell that can be ameliorated by decreasing the number of repeats (*Salim et al., 2017*).

We determined rDNA size by assessing the fraction of DNA sequencing reads from a sonicated G1 population that arose from the rDNA. While deletion of *FUN30* had no notable effect on rDNA size in a *SIR2* strain, it caused a 50% reduction in a *sir2* mutant, from 9.2% to 4.2% (*Figure 2C*). rDNA size reduction of a similar magnitude was observed in the ATPase-deficient Fun30K603R mutant, but not in the non-phosphorylatable Fun30S20AS28A mutant, mirroring the disparate effect of the two alleles on rDNA replication timing. The reduction of the rDNA size in *sir2 fun30* mutants was confirmed by qPCR analysis of DNA isolated from log cultures, and by Southern blots (*Figure 3*, *Figure 2—figure supplement 1*). Based on measurements in log-growing cells, we estimated the rDNA size in *fun30 sir2* mutants to be ~40 repeats (*Figure 2—figure supplement 1*). This places it in a category comparable to the shortest rDNA arrays in a collection of DNA replication mutants noted for their contraction of the rDNA (*Salim et al., 2017*), thus emphasizing the severity of the replicative disadvantage conferred by the deletion of *FUN30* when Sir2 function is absent.

Alterations of rDNA size are promoted by the replication fork barrier (RFB), which imposes a unidirectional replication fork arrest to prevent head-to-head collisions between DNA polymerase and RNA polymerase I while transcribing 35S units. Replication forks that are stalled at the RFB are prone to breakage, which can lead to unequal sister chromatid exchange and fluctuations in rDNA array size. These rDNA size fluctuations are exacerbated in *sir2* mutants, which exhibit an increased rate of recombination, resulting in an unstable rDNA array size. Deletion of *FOB1*, which abolishes RFB function, stabilizes rDNA size by reducing, though not eliminating, the frequency of recombination-initiating double-strand breaks.

We have interpreted the shrinkage of the rDNA caused by deletion of *SIR2* and *FUN30* to be a specific example of the general phenomenon demonstrated by *Salim et al., 2017*, i.e., that mutations that impair replication generate selective pressure that favors rDNA shrinkage. According to this interpretation, while deletion of *FOB1* is expected to retard the transition to reduced array sizes, it should not prevent it, and thus rDNA shrinkage should not be *FOB1*-dependent.

To determine whether the reduction of rDNA size in *sir2 fun30* mutants was dependent on RFB activity and *FOB1*, we deleted *FUN30* in a *sir2 fob1* strain with 160 repeats. Using this strategy, we obtained *fob1 sir2 fun30* transformants with over 150 rDNA repeats. If rDNA shrinkage does not require Fob1, it should occur, albeit slowly, in *fob1* mutants. To assess this, we continuously grew several independent *fob1 sir2 fun30* colonies, along with control *fob1 sir2*, and *fob1 fun30* strains for 120 generations. We found that, unlike the control *fob1* strains, which maintained a stable rDNA size, all four *fob1 sir2 fun30* cultures reduced their rDNA arrays size to fewer than 100 copies by 120 division (*Figure 2—figure supplement 2*). These results demonstrate that, although *FOB1* affects the kinetics of rDNA size reduction in *sir2 fun30* strains, the reduced rDNA array size upon *FUN30* deletion size does not depend on *FOB1*. Thus, suppressed rDNA replication drives the selection for reduced rDNA array size in both RFB-proficient and RFB-deficient *fun30 sir2* cells. Consistent with this notion, our S-seq results showed delayed rDNA replication timing in *fob1 sir2 fun30* cells with 150 rDNA repeats, compared to their *fob1 sir2* counterparts (*Figure 2—figure supplement 3*). In summary, we conclude that Fun30 nucleosome-remodeling activity promotes rDNA replication and is required for maintaining rDNA size in *sir2* mutants. We next aimed to determine whether Fun30 exerts this effect by reducing MCM loading or the ability of loaded MCMs to fire.

## High level of origin licensing at rDNA origins in WT, *sir2*, and *sir2 fun30* mutants

It has been estimated that ~20% of potential rDNA origins are activated during any single S phase in WT cells (*Brewer and Fangman, 1988*; *Linskens and Huberman, 1988*). Furthermore, single-molecule studies suggested that the fraction of active origins is increased in *sir2* mutant cells (*Pasero et al., 2002*; *Yoshida et al., 2014*). To determine whether *SIR2* and *FUN30* modulate rDNA replication by changing origin licensing, i.e., the fraction of origins that are bound by MCM helicase complexes prior to the beginning of S phase, we assessed licensing in WT, *sir2*, *fun30*, and *sir2 fun30* cells by measuring DNA cleavage by MCM2-MNase using Southern blots (*Foss et al., 2021*): Digestion with XmnI yields a 3.5 kb fragment that spans the rARS (*Figure 3*), and, if the rARS is licensed, activation of MNase by addition of calcium will eliminate this fragment, thus making it undetectable when using a probe upstream of the rARS. As a control, we used a fragment that, based on our MCM-ChEC data, should not contain MCM complexes. Activation of MNase in all four genotypes led to an ~60–85% decrease in the XmnI fragment (*Figure 3*, *Figure 3—figure supplement 1*), therefore we conclude that (1) licensing is high, at ~60–85%; and (2) a much higher fraction of origins are licensed than are activated, making it unlikely that significant regulation of origin activity occurs at the level of licensing. This is the first quantitative assessment of licensing at the rDNA in either WT or *sir2*.

## Fun30 promotes firing of rDNA origins and suppresses firing of non-rDNA origins in *sir2* mutants

Our observation that *FUN30* status had no obvious effect on rDNA origin licensing suggested that Fun30 might instead advance rDNA replication in sir2 mutants by promoting rDNA origin firing. To evaluate this possibility, we measured the effect of *FUN30* status on origin activity at the rDNA using 2D gels. We grew cells to log phase, arrested them for 1.5 hr in alpha factor and collected them at different time points after release into 200 mM hydroxyurea (HU). Prior to separation on two-dimensional (2D) gels, DNA was digested with NheI, which releases a 4.7 kb rARS-containing linear DNA fragment at the internal rDNA repeats (1N) and a much larger, 24.4 kb single-rARS-containing fragment originating from the rightmost repeat (*Figure 4A*). In 2D gels, active origins generate replication bubble arc signals, whereas passive replication of an origin appears as a y-arc (*Figure 4B*). Having a signal emanating from a single-rARS-containing fragment (*Figure 4A and B*) simplifies the comparison of rDNA origin activity in strains with different numbers of rDNA repeats, such as in *sir2* vs *sir2 fun30* mutants. Origin activity is expressed as a ratio of the bubble to the single-ARS signal, effectively measuring the number of active rDNA origins per cell at a given time point.

As seen previously, deletion of *SIR2* increased the number of activated rDNA origins, while deletion of *FUN30* suppressed this effect (*Figure 4D*). When analyzed in aggregate at 20, 30, 60, and 90 min following release into HU, the average number of activated rDNA origins in *sir2* mutant was increased 6.3-fold compared to those in WT (5.0±2.3 in *sir2* vs 0.8±0.4 in WT, p<0.05 by two-tailed t-test), and this increased number was reduced upon *FUN30* deletion (1.3±0.7 in *sir2 fun30*, p<0.05 by two-tailed t-test vs *sir2*, NS for comparison to WT). Deletion of *FUN30* in a strain with a WT copy of *SIR2* did not have a significant effect. We conclude that deletion of *FUN30* partially reverses the accelerated activation of ribosomal origins caused by deletion of *SIR2*.

The overall impact of *FUN30* deletion on rDNA origin activity in a *sir2* background is expected to be a composite of two opposing effects: a suppression of rDNA origin activation and increased rDNA origin activation due to reduced rDNA size. To evaluate the effect of *FUN30* on rDNA origin activation independently of rDNA size, we generated an isogenic set of strains in a *fob1* background, all of which contain ~30 copies of the rDNA repeats. (Deletion of *FOB1* is necessary to stabilize rDNA copy number.) Comparing rDNA origin activity in *sir2* vs *sir2 fun30* genotypes, we observed a robust and reproducible reduction in rDNA origin activity upon *FUN30* deletion (*Figure 4C*). This finding confirms that the *FUN30* suppresses rDNA origin firing in *sir2* background independently of both rDNA size and *FOB1* status.

We and others have previously shown that deletion of *SIR2* not only advances replication at the rDNA but also retards replication elsewhere in the genome (*Kwan et al., 2013*; *Yoshida et al., 2014*). The absence of *SIR2* has been linked to decreased activation of less robust early replication origins, likely due to their heightened sensitivity to overall levels of replication-initiating factors. To ascertain the specificity of the interaction between *FUN30* and *SIR2*, we investigated whether deleting *FUN30*

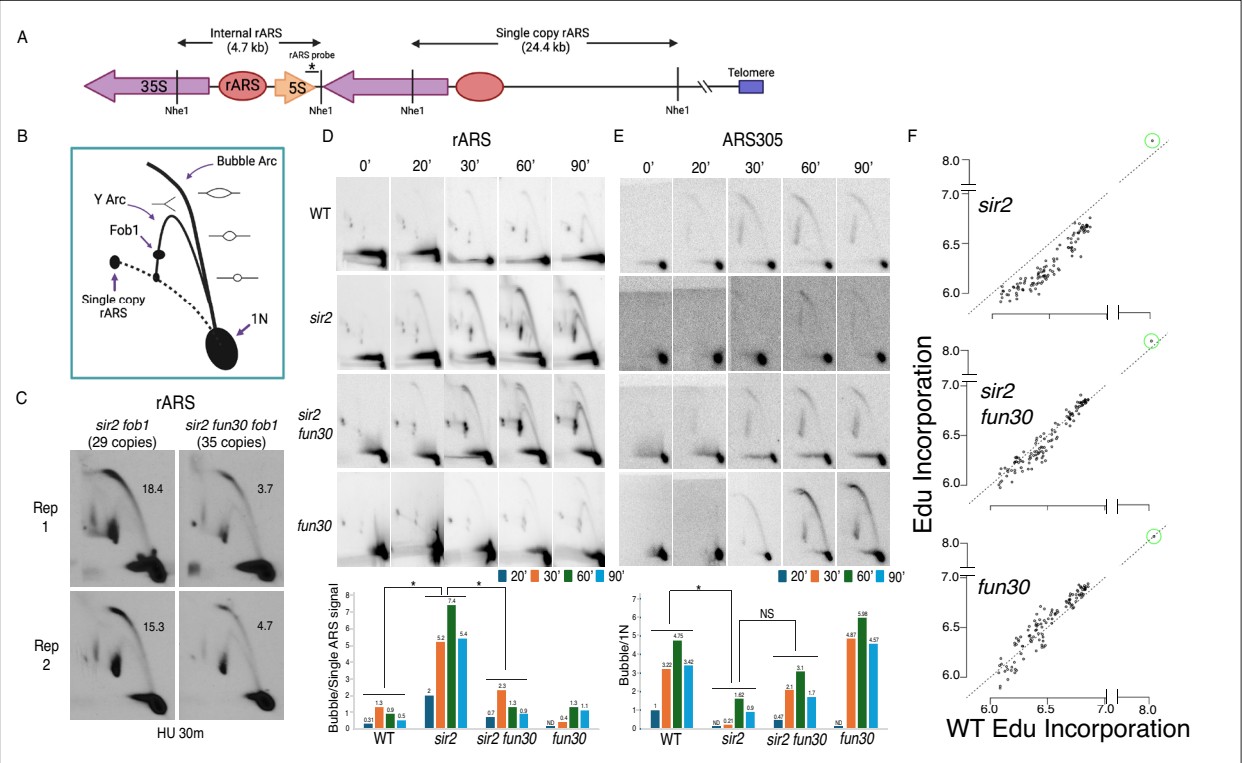

**Figure 4.** Deletion of *FUN30* suppresses origin activity at ribosomal DNA (rDNA) and promotes it elsewhere in the genome. (**A**) Schematic of the right end of the rDNA locus depicting Nhe1 cut sites. Digestion with Nhe1 releases multiple copies of a 4.7 kb rARS-containing internal fragment (1N) and a single copy of a 24.4 kb fragment that contains the rightmost rDNA origin. (**B**) Diagram of replication structures detected by two-dimensional (2D) gel. (**C**) Replicas of 2D gels showing replication at the rDNA locus 30 min after release from alpha factor into hydroxyurea (HU) in *sir2 fob1* (17564) or *sir2 fun30 fob1* (17556) strains with short (29 copies and 35 copies, respectively) rDNA arrays. Numbers indicate the ratio of bubble arc to single copy ARS signal, reflecting the number of activate origins per cell. The average number of activate origins (mean ± SD) for *sir2 fob1* is 16.9±2.2 vs 4.2±0.7 (p<0.05 by t-test). (**D**) 2D gels showing replication at the rDNA. Cells were arrested in G1 with alpha factor and then released into medium containing 200 mM HU. Quantitation of the ratio of bubble arc to 24.4 kb single-copy rARS is shown below. * indicates significant differences (p≤0.05 as determined by comparing combined signals from all four time points by Student's t-test). Strain numbers used were as follows: wild-type (16747), *sir2* (16769), *sir2 fun30* (17257), and *fun30* (17256). (**E**) Replication of ARS305 was examined as in D, but the 1N spot was used for normalization and ratios of bubble arc to 1N spot were normalized to this ratio for the 20 min time point in wild-type. (**F**) Ethynyl-deoxy-uridine (EdU) incorporation at 111 early origins 1 hr after release from G1 into medium containing 200 mM HU. Total genome-wide read counts in each sample were normalized to the number of reads in the sample with the highest total, thereby normalizing numbers to genome-wide incorporation of EdU. Each dot represents a single origin, with read depths summed across a 5 kb window centered on the MCM binding site within each origin. Points are plotted according to EdU signal in wild-type on the x-axis and in the mutant in question on the y-axis. Suppression of origin activity is reflected in points dropping below the dotted line at 45°. The rARS is circled in green. Strain numbers used were as follows: wild-type (17265), *sir2* (17271), *sir2 fun30* (17279), and *fun30* (17281).

The online version of this article includes the following source data for figure 4:

**Source data 1.** PDF file containing original Southern blots displayed in *Figure 4C* indicating the relevant bands and treatments.

**Source data 2.** Original files for Southern blot analysis displayed in *Figure 4C*.

**Source data 3.** Calculations of bubble to single-ARS ratios shown in *Figure 4C*.

**Source data 4.** PDF file containing original Southern blots of two-dimensional (2D) gels displayed in *Figure 4C and D* indicating the relevant genotypes and time points for rARS and ARS305.

**Source data 5.** Original files for Southern blot analysis displayed in *Figure 4C and D*.

**Source data 6.** Calculations of bubble to 1N ratios plotted in bar graphs in *Figure 4C and D*.

**Source data 7.** Source data for plots in *Figure 4F* and statistical analysis of MCM abundance.

also mitigates the delayed activation of weak early origins in a *sir2* mutant background. We focused on the activation of ARS305, an origin we previously identified as delayed in a *sir2* mutant using 2D gel analysis. Overall, the comparison of origin activity at ARS305, measured as bubble arc to 1N signal, with that at rDNA demonstrates a reciprocal relationship between activity at ARS305 and

rDNA across the four genotypes (*Figure 4E*). As seen previously, deletion of *SIR2* suppresses ARS305, whereas additional deletion of *FUN30* increased activity close to that of WT. Deletion of *FUN30* in a *SIR2* background also showed a trend of increased origin activation, though this difference was not statistically significant.

Delayed initiation of genomic origins following *SIR2* deletion has been previously demonstrated on a global scale as a decline in the incorporation of a labeled uridine analogue, bromodeoxyuridine, at weak early origins in *sir2* cells compared to WT cells upon their release from G1 into HU (*Yoshida et al., 2014*). We employed a similar approach to investigate whether the deletion of *FUN30* mitigates this effect, utilizing a different labeled nucleotide, ethynyl-deoxy-uridine (EdU), which can be isolated through click chemistry rather than immunoprecipitation. WT, *sir2*, *sir2 fun30*, and *fun30* cells engineered for the uptake and phosphorylation of the analogue (*Viggiani and Aparicio, 2006*) were arrested in G1 using alpha factor and collected 60 min after release into medium containing HU and EdU. DNA extracted from these cells was sonicated and subjected to click chemistry with biotin azide to covalently link biotin to EdU incorporated into DNA. Biotinylated DNA was then purified and sequenced. The impact of *SIR2* and *FUN30* status on replication timing was assessed by graphing the EdU signal at 111 early origins in WT cells on the x-axis against the signal in *sir2*, *fun30*, and *sir2 fun30* cells on the y-axis (*Figure 4F*). Similar to observations with BrdU, we noted reduced EdU incorporation at a subset of early origins in *sir2* mutants, coupled with increased EdU incorporation at rDNA origins. Deletion of *FUN30* in a *sir2* background partially restored EdU incorporation at early origins, concomitant with reduced EdU incorporation at rDNA origins. In particular, the median value of $\log_{10}$ of read depths at 111 early origins, as the data shown in *Figure 4F*, dropped from 6.5 for WT to 6.2 for *sir2* but then returned almost to WT levels (6.4) in *sir2 fun30*. The p-value obtained by Student's t test, comparing the drop in 111 origins from WT to *sir2* with that from WT to *sir2 fun30* was highly significant ($<<10^{-16}$). In contrast, *FUN30* deletion in the WT background did not reduce EdU incorporation at genomic origins (median 6.6). These findings highlight that *FUN30* deletion-induced suppression of rDNA origins in *sir2* is accompanied by the activation of genomic origins.

## Early firing of displaced helicase complexes

While our results above demonstrate that deletion of *SIR2* promotes firing of rDNA origins and that additional deletion of *FUN30* suppresses this phenotype, they lack the resolution to distinguish replication arising from the displaced vs the nondisplaced MCM complexes. Other methods for measuring origin activation, such as generation of single-stranded DNA or incorporation of nucleoside analogs, also suffer from this limitation. Distinguishing between firing of the displaced vs the nondisplaced MCM complexes is important to test our hypothesis that displacement of the helicase complex into a relatively nucleosome-free region is responsible for its early activation. We therefore sought to devise a method that can differentiate activation of MCM complexes that are only ~150 bp apart, i.e., the approximate distance between displaced and nondisplaced MCMs. We reasoned that, because MCM-ChEC has the resolution to distinguish closely spaced helicase complexes, the differential disappearance of such closely spaced footprints might permit us to distinguish their relative activation times.

To explore the possibility that the disappearance of the MCM-ChEC signal could be used to distinguish the firing of the displaced vs nondisplaced populations of MCM, we first asked whether the pattern of disappearance of the MCM signal recapitulates that of origin firing. In particular, while late origins are inhibited by HU, early origins are not; therefore, we asked whether this phenomenon was reflected in MCM-ChEC. To do this, we arrested cells in G1 using alpha factor and then released them into medium containing HU, taking samples for ChEC analysis at 15, 30, 45, 60, and 90 min (*Figure 5A*). HU inhibits ribonucleotide reductase and thereby dramatically slows S phase, making it easier to follow replication kinetics, but the relative order of firing of different origins remains unchanged. In *Figure 5B*, we plot the MCM-ChEC signal at 111 early origins and 101 late origins, with G1 on the x-axis and S phase on the y-axis. While we observed little difference in the signal abundance between early and late origins just 15 min after release from alpha factor, before origins have had a chance to fire, this difference increases over the time course, with early origins dropping more than late. MCM signal quantification demonstrates that the differences in MCM abundance between early and late origins increase over the time course in HU (*Figure 5—figure supplement 1*). Using chromatin immunoprecipitation followed by sequencing (ChIP-seq) for FLAG-tagged MCM2 in cells arrested in alpha factor and those released into HU for 60 min, we observed a similar reduction of the MCM signal at

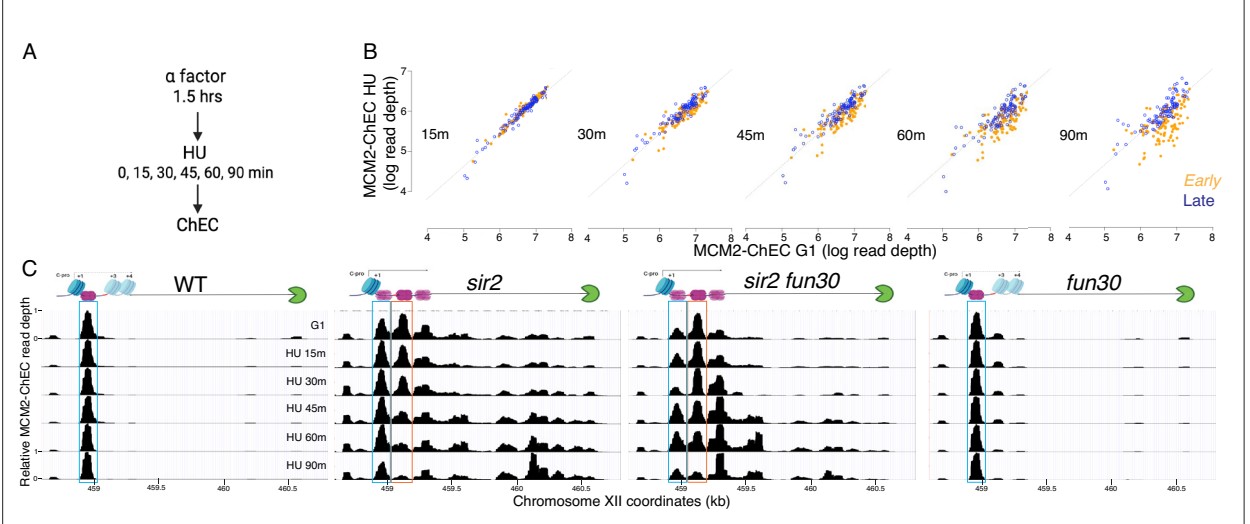

**Figure 5.** Disappearance of MCM2-chromatin endogenous cleavage (ChEC) signal can be used to monitor origin firing. (**A**) Cells were synchronized in G1 phase using alpha factor for 1.5 hr before being released into media supplemented with hydroxyurea (HU). Cells were harvested at various time points post-release (15–90 min) and subjected to MCM2-ChEC analysis. (**B**) Decrease in MCM2-ChEC signal in wild-type (16747) for 111 early (orange) and 101 late (blue) origins. MCM2-ChEC signal was quantified over 200 base pair windows centered on the MCM binding site within each origin. Each point is plotted according to MCM2-ChEC signal at the time point in question on the y-axis and the corresponding signal in G1 on the x-axis, thus decrease in signal appears as a drop below the 45° diagonal. (**C**) MCM signal at the ribosomal DNA (rDNA). MCM2-ChEC signal in G1 appears predominantly at the location indicated by the blue rectangle seen in wild-type (16747) and *fun30* (17256), whereas it is spread across both the blue and orange rectangles in the absence of Sir2. The displaced MCM2-ChEC signal (orange rectangle) in *sir2* (16769) disappears more rapidly than its nondisplaced counterpart, and this effect is suppressed by *fun30* (17257). Decrease of MCM2-ChEC signal at the rARS is accompanied by increased signal to the right.

The online version of this article includes the following source data and figure supplement(s) for figure 5:

**Source data 1.** Source data for plots in *Figure 5B*, *Figure 5—figure supplement 1*.

**Source data 2.** Source data for plots in *Figure 5C*.

**Figure supplement 1.** Quantification of MCM2-chromatin endogenous cleavage (ChEC) signal at 111 early (orange) and 101 late (blue) origins as cells progress through S phase (quantitation of *Figure 5B*).

**Figure supplement 1—source data 1.** Statistical analysis of data plotted in *Figure 5—figure supplement 1*.

**Figure supplement 2.** Quantitation of early and late origins using MCM2-chromatin immunoprecipitation (ChIP).

**Figure supplement 2—source data 1.** Source data for plots in *Figure 5—figure supplement 2*.

**Figure supplement 3.** Change in localization of MCM2-chromatin immunoprecipitation (ChIP) signal at the ribosomal DNA (rDNA) with progression from G1 into S phase.

**Figure supplement 4.** Deletion of *FUN30* suppresses activation of the displaced MCM complex at the ribosomal DNA (rDNA) (replica of results in *Figure 5C*).

**Figure supplement 4—source data 1.** Source data for plots in *Figure 5—figure supplement 4*.

**Figure supplement 5.** Displaced MCM complex is activated early in wild-type (WT).

**Figure supplement 6.** C-pro transcript levels.

**Figure supplement 6—source data 1.** Source data for plots in *Figure 5—figure supplement 6*.

early compared to later origins, confirming our MCM-ChEC results (*Figure 5—figure supplement 2*). We conclude that the disappearance of the MCM-ChEC signal as cells progress through S phase can be used to assess origin firing.

We then turned our attention to the MCM signals at the rDNA origin; here the presence of the replication fork block approximately 1.5 kb from the sites of MCM binding provides a fortuitous means of confirming our interpretation, since the loss of the MCM-MNase signal at the origin should appear as a reciprocal increase where the replisome stalls. We first used MCM2-Flag ChIP in WT and *sir2* cells after release into HU to confirm this prediction. In both WT and *sir2* cells, the MCM can be found at the rDNA origin in alpha factor-arrested cells as expected (*Figure 5—figure supplement 3*).

However, at 60 min following release into HU, unlike in WT cells where the MCM signal remains at its loading site, in *sir2* mutants, we can see that a large fraction of the signal has now redistributed from its loading site and accumulated left of the RFB, consistent with early activation of rDNA origins in *sir2* cells and the arrest of the replication forks at the RFB. While these results demonstrate that ChIP can be used to monitor the repositioning of the MCM signal at a kb scale and origin activation, its resolution is inadequate for small-scale changes in MCM signal distribution, which is needed to monitor the proposed differences in activation of repositioned and non-repositioned MCM complexes. We therefore turned to ChEC to address this question.

The MCM-ChEC signal in WT cells remained at their initial loading throughout the duration of the HU experiment (blue box in *Figure 5C*) as we have seen by ChIP in WT cells at 60 min. (Note that 200 mM HU causes an approximately 20-fold slowing of S phase, therefore a 60 min time point is equivalent to <5 min without HU, i.e. prior to the time when the rDNA origin fires in WT cells.) Like in WT, this nondisplaced population of MCM in *sir2* remains relatively constant (blue box); however, in striking contrast, the displaced population of MCM in *sir2* (orange box) shows a continual decrease over the course of the experiment, and there is a concomitant increase in signal in the region upstream of RFB that is absent in WT. Furthermore, both the decrease in the displaced signal (orange box) and its corresponding increase upstream of the RFB were suppressed by deletion of *FUN30*. The changes in the relative abundance of displaced and nondisplaced MCMs during the time course, as well as the repositioning of the MCM signal to the region upstream of RFB across different genotypes, were highly reproducible (*Figure 5—figure supplement 4*).

Taken together, these observations provide strong support for the hypothesis that the displaced population of MCM in *sir2* cells fires more readily than its nondisplaced counterpart and that the increased firing of the displaced MCMs is suppressed by *FUN30* deletion.

It is notable that, although a small fraction of the total G1 signal in WT arises from the displaced location, this minor signal likewise disappears more rapidly than its nondisplaced counterpart (*Figure 5—figure supplement 5*). This suggests that some feature of the immediate chromatin environment may be responsible for the difference in firing times at the two locations, and that this difference is present in both WT and *sir2*; in other words, the fact that the displaced helicase complex is activated before the nondisplaced complex regardless of *SIR2* status is consistent with a model in which *sir2* mutants replicate their rDNA early by virtue of displacing much of their loaded helicases to a region where they are more apt to fire, rather than by altering the chromatin environment more globally to affect all replicative helicases in the nucleolus.

Finally, although deletion of *FUN30* could suppress replication initiation at the rDNA either by inhibiting the firing of the active, repositioned MCM complex, or by preventing MCM repositioning to the 'active location' in the first place, our results suggest that suppression occurs through the former mechanism. Consistent with previous reports that *fun30* mutants are deficient in transcriptional silencing (*Neves-Costa et al., 2009*), C-pro RNA levels were approximately twice as high in *sir2 fun30* cells compared to *sir2* cells when adjusted for rDNA size (*Figure 5—figure supplement 6*).

Moreover, deletion of *FUN30* shifts the distribution toward the repositioned MCM location over the non-repositioned one in G1 cells (*Figure 5C*), aligning with the increased C-pro transcription observed in *fun30* mutants. This shift is evident in both *sir2* and *SIR2* cells. Despite the increased transcription-mediated repositioning in *sir2 fun30* cells compared to *sir2* cells during G1, repositioned MCM persists longer in *sir2 fun30* cells than in *sir2* cells after release into HU. Additionally, *sir2 fun30* mutants exhibit reduced MCM accumulation at the RFB compared to *sir2* mutants after release into HU, supporting the conclusion that MCM disappearance in HU reflects origin activation rather than transcription-mediated displacement.

## Fun30 maintains low nucleosome occupancy near repositioned MCMs

Our findings demonstrate that MCMs, when situated at their loading site adjacent to a high-occupancy +1 nucleosome, exhibit reduced firing propensity. In contrast, when these MCMs are displaced to a neighboring region with lower nucleosome occupancy, their firing propensity significantly increases. This heightened firing tendency is dependent on Fun30 CRE and its remodeling activity. This suggests that the absence of *FUN30* could elevate nucleosome occupancy around the relocated MCMs, consequently inhibiting their ability to fire.

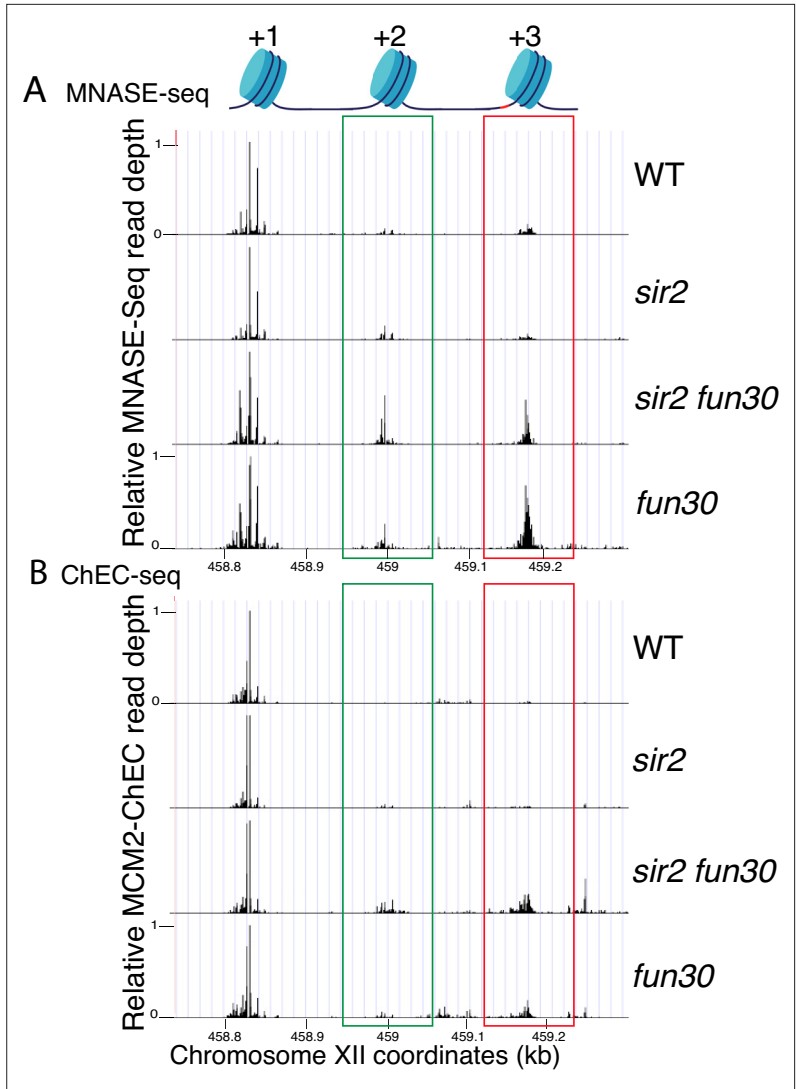

**Figure 6.** Nucleosome occupancy assessed by micrococcal nuclease (MNase)-seq and MCM2-chromatin endogenous cleavage (ChEC). (**A**) Analysis of nucleosome occupancy at ribosomal DNA (rDNA) origins using MNase-seq revealed a consistent high occupancy of the +1 nucleosome across wild-type (WT) (16747), sir2 (16769), sir2 fun30 (17257), and fun30 (17256), which served as our normalization reference. Occupancy at the +2 (green box) and +3 (red box) positions was increased by deletion of *FUN30* in both sir2 and *SIR2*. MCM2-ChEC signal was quantified specifically from the 151–200 base pair (nucleosome) size range. (**B**) Analysis of nucleosome occupancy with MCM2-ChEC (see **Figure 1B**) reveals nucleosome occupancy in that subset of cells and rDNA repeats in which MCM is present. Deletion of *FUN30* leads to increased occupancy at the +2 and +3 positions in a sir2 background. MCM2-ChEC signal was quantified specifically from the 151–200 base pair (nucleosome) size range.

The online version of this article includes the following source data and figure supplement(s) for figure 6:

**Source data 1.** Source data for plots in *Figure 6*.

**Figure supplement 1.** Nucleosome occupancy assessed by micrococcal nuclease (MNase)-seq and MCM2-chromatin endogenous cleavage (ChEC) (replica of *Figure 6*).

**Figure supplement 1—source data 1.** Source data for plots in *Figure 6—figure supplement 1*.

To evaluate this possibility, we examined the effect of *FUN30* deletion on nucleosome occupancy at rDNA origins in WT and *sir2* background using MNase-seq in G1-arrested cells. Chromatin treatment with MNase digests DNA that is not bound by proteins, such as nucleosomes and transcription factors. Nucleosomes, which are wrapped by approximately 150 base pairs of DNA, tend to protect fragments in the 150–200 base pair range, and thus profiling fragments in this size range are

widely used to assess nucleosome distribution. Plotting nucleosome midpoints at the rARS in G1 cells (*Figure 6A*, *Figure 6—figure supplement 1*) reveals that nucleosome occupancy at the +2 (green box) and +3 (red box) positions is higher in the *sir2 fun30* double mutant than in the *sir2* single mutant. Furthermore, this effect of Fun30 is not limited to the *sir2* background, since nucleosome occupancy is also modestly increased in *fun30* cells compared to WT cells. These observations support the notion that it is the relatively nucleosome-free nature of the region into which MCMs are displaced in *sir2* that make the displaced complexes more prone to activation. Furthermore, this suggests that it could be particularly informative to assess nucleosome occupancy specifically adjacent to loaded MCM complexes, as described below.

It has previously been observed, both with MCM and with various transcription factors, that the ChEC technique can reveal not only the binding site of the protein to which the MNase is fused, but also, if present, that of the adjacent nucleosome (*Zentner et al., 2015*; *Foss et al., 2019*). This is due to the fact that the entrance and exit sites for DNA wrapped around a nucleosome are in close physical proximity, despite the fact that they are separated by approximately 150 base pairs, and thus the MNase fusion protein adjacent to a nucleosome often cleaves both sites. MCM-ChEC, therefore, provides a powerful tool for measuring nucleosome occupancy immediately adjacent to the MCM helicase complex *exclusively in those rDNA repeats where those complexes have been loaded*. By plotting the midpoints of 150–200 bp fragments released by MCM2-MNase, we were able to determine that nucleosome occupancy at both the +2 and +3 nucleosomes, i.e., those adjacent to the displaced MCM complexes, are elevated in *sir2 fun30* as compared to *sir2* (*Figure 6B*, *Figure 6—figure supplement 1*). This finding indicates that Fun30 is required for maintaining the low nucleosome occupancy in the region into which the preponderance of MCM complexes have been displaced in *sir2* mutant cells.

## Discussion

In this study, we have attempted to shed light on the mechanism by which Sir2 represses replication origins at the rDNA. Sir2's activity has been widely ascribed to the intuitive, but mechanistically imprecise, notion that 'closed chromatin' is less accessible to both transcription and replication factors. Consistent with this model, deletion of *SIR2* abolishes heterochromatic compaction and simultaneously increases the ability of methylases and restriction endonucleases to act on those sequences (*Gottschling, 1992*; *Singh and Klar, 1992*; *Loo and Rine, 1994*; *Weiss and Simpson, 1998*; *Ansari and Gartenberg, 1999*; *Ravindra et al., 1999*). However, at least in its simplest form, non-specific steric exclusion of DNA metabolic enzymes cannot fully account for the suppression of transcription and replication at the rDNA. Steric arguments such as these are even less compelling when made for rDNA than for the silent mating type loci and telomeres, because chromatin compaction has been studied mostly in the context of the complete Sir complex (Sir2–4) (*Gartenberg and Smith, 2016*). In contrast, Sir3 and Sir4 are not present at the rDNA (*Straight et al., 1999*). Moreover, rDNA is the most highly transcribed region of the genome, with transcription by PolI and PolIII accounting for approximately 80% of cellular RNA (*Warner, 1999*; *Woolford and Baserga, 2013*). These two polymerase complexes are similar in size to the PolII complex, thus size alone is not sufficient to selectively occlude PolII.

Additionally, deletion of SIR2 increases accessibility of restriction enzymes to heterochromatin at the HML locus by 1.5-fold, yet it increases PolII transcription by over 3000-fold (*Holland, 2002*). This indicates that the magnitude of Sir2's steric exclusion is insufficient to explain its effect on PolII transcription. Similar measurements in eukaryotic cells found little difference in accessibility of restriction enzymes and MNase between euchromatin and heterochromatin (*Chereji et al., 2019*).

In the current report, we argue that Sir2's role in suppressing replication at the rDNA reflects local differences in the distribution of the MCM replicative helicase complexes rather than more global occlusion of these complexes from the rDNA. In particular, our results suggest that the key difference between WT and *sir2*, with regard to replication initiation, is the location of the MCM complex, with the preponderance of that complex in WT abutting a high-occupancy nucleosome while, in *sir2*, it is mostly in a relatively nucleosome-free region. Furthermore, we show that, even though this nucleosome-free region is less than 200 base pairs from MCM's original location, the displaced helicase complex is significantly more prone to activation than its nondisplaced counterpart. Moreover, the chromatin-remodeling enzyme Fun30 is required to maintain the nucleosome-free character

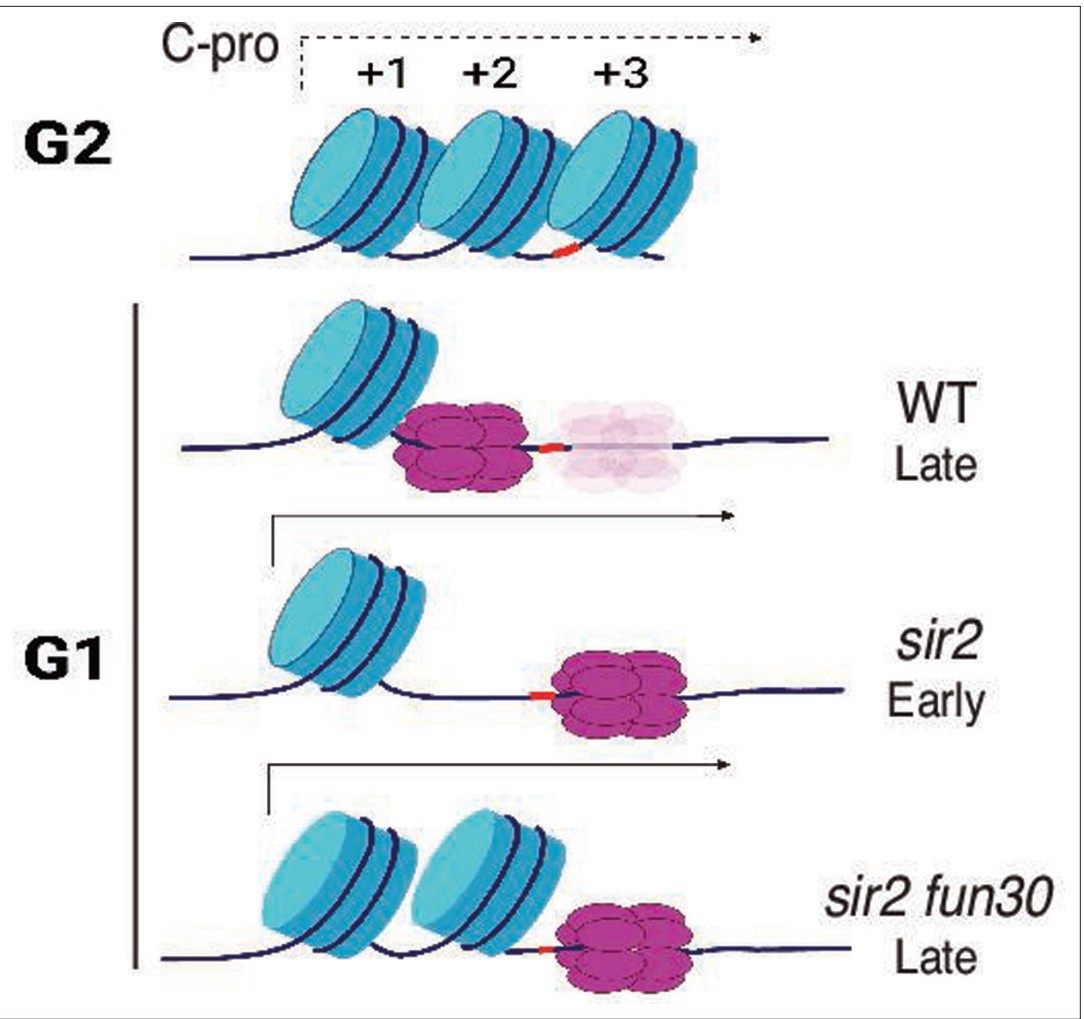

**Figure 7.** Model for relationship between MCM location and replication timing at the ribosomal DNA (rDNA). MCM helicase complex (purple ovals) in wild-type abuts the +1 nucleosome (blue cylinder) in G1, making it relatively resistant to activation. Deletion of *SIR2* de-represses C-pro transcription (arrow pointing to the right), and RNA PolII pushes MCM complex to a nucleosome-free area, where it is more prone to activation. Deletion of *FUN30* in a *sir2* mutant leads to increased nucleosome occupancy at the +2 position, adjacent to MCM complex, making this complex resistant to activation. Short red stretch of DNA (e.g. between +2 and +3 nucleosomes in top row) indicates ARS consensus sequence (ACS). 'Ghost' Mcm complex in wild-type indicates the small fraction of Mcm that is displaced even in wild-type cells (see *Figure 5—figure supplement 5*). Created with BioRender.com.

around the displaced MCM, and its deletion both increases nucleosome occupancy and decreases the ability of the displaced MCM to fire. Finally, as discussed below, we suggest that the increased proclivity of the displaced MCM to fire is not an anomaly specific to the physiology of *sir2* cells, but is an integral feature of replication of the rDNA during normal S phase.

Our model for the manner in which Sir2 regulates rDNA replication depends critically on the notion that the local nucleosome landscape can influence the activation of MCM helicase complexes (*Figure 7*), and in vitro experiments suggest that this is, indeed, the case *Azmi et al., 2017* monitored both licensing and firing in vitro, using a 3.8 kb nucleosomal template containing the ARS1 origin. They found that, while licensing was equivalent whether the template had been populated with nucleosomes using Swi/Snf or Rsc remodelers, on the one hand, or Isw1a, on the other, the loaded MCM complexes in the latter templates were more prone to activation. Furthermore, MCM complexes on Swi/Snf- or Rsc-remodeled templates could be made more prone to activation if subsequently treated with Isw1a, although the converse was not true. The salient difference between the two types of nucleosomal landscapes, however, was not obvious, as the distance

between the MCM complex and the nearest nucleosome was not notably affected, although it is possible that the composition (e.g. H2A/H2B content) differs. It was also unclear how the nucleosome landscape affected activation of MCM complexes, though the relevant step in activation was determined to be the formation of the CMG complex with Cdc45 and GINS rather than the prior phosphorylation of the complex by DDK. In summary, while it is clear that the local nucleosome landscape can affect origin activity (*Eaton et al., 2010*; *Belsky et al., 2015*), the underlying mechanism remains murky.

Regardless of the reason for the increased proclivity of the displaced MCM to fire, the observation has implications for replication at the rDNA not only in *sir2* mutants but also in WT. We have previously shown that the rightward displacement of MCM double-hexamers in *sir2* is due to de-repression of C-pro transcription (*Foss et al., 2019*), while our current results demonstrate that the displaced MCM is more prone to activation. Combining these two observations, we suggest that the rDNA replicates earlier in *sir2* than in WT simply because more of the MCM complex is in a location that favors its activation; in other words, we favor the view that the region of relatively low nucleosome occupancy to the right of MCM's normal location is more amenable to MCM activation, regardless of *SIR2* status. In support of this notion, close analysis of our MCM-ChEC data reveals that, while MCM complexes at the displaced location are much rarer in WT than they are in *sir2*, this particular subpopulation appears equally susceptible to activation in both genotypes. This has the important implication that the manner in which the rDNA origin fires in *sir2* mutants may be simply an exaggerated instance of its normal behavior, and that even in WT cells, it is predominantly from MCMs that are displaced, likely due to low level of C-pro transcription, that replication initiates.

Classic studies that employed EM and 2D gel electrophoresis showed that only a fraction of rDNA origins, estimated at one in five, are activated in any single cell cycle in yeast (*Brewer and Fangman, 1988*; *Linskens and Huberman, 1988*). These estimates were later corroborated by single molecule analysis, which also observed that the fraction of active origins increases in *sir2* mutants (*Pasero et al., 2002*). This low fraction of active rDNA origins could be due to insufficient origins licensing, suppression of activation or licensed origins, or a combination of the two mechanisms. Our measurements of rDNA licensing showed that MCM are loaded in up to 90% of origins in WT cells and allowed us to conclude that regulation or rDNA replication initiation occurs primarily at the level of origin firing.

CREs have been proposed to assist DNA replication by modulation of nucleosome phasing in the vicinity of replication origins, but it has proven difficult to assign specific roles to remodelers in genome replication in vivo (*Cutler et al., 2018*). One of the reasons for this difficulty is the redundancy among different remodelers on one hand, and the multiple essential roles CRE play in DNA transactions in addition to replication. For example, in vitro studies have identified four different remodelers that in combination with ORC can phase nucleosome at 320 replication origins (*Chacin et al., 2023*). However, a deletion of any single of these remodelers had no discernible effect on nucleosome phasing in vivo and a quadruple mutant was required to abolish phasing. While the characterization of this quadruple mutant suggested impaired replication initiation, it was difficult to exclude indirect effects that might have been caused by the effects on transcription. In contrast, the effect of a chromatin remodeler on DNA replication we describe in this work is remarkably specific: we found that the activity of a single remodeler, Fun30, is required to activation not only of a specific origin, but that its action is specific to nucleosomes adjacent to repositioned MCM complexes at the rDNA. The suppression of rDNA origin activity in sir2 mutants, which contained a large fraction of repositioned MCMs, by *FUN30* deletion is so severe that it reduces the rDNA size ~3- to 5-fold, from 160 to 35–50 copies.

In conclusion, our findings contribute to a growing body of evidence highlighting the importance of positioning of the MCM replicative helicase within the local nucleosome environment in determining replication timing (*Azmi et al., 2017*; *Rodriguez et al., 2017*). While it is clear that small differences in this positioning can have significant effects on the timing of DNA replication, it is unlikely that a simple rule exists to directly infer origin timing solely based on the distance between the MCM complex and the nearest nucleosome. Nonetheless, our results suggest that this parameter could be a crucial piece of the puzzle in understanding why some origins fire early in S phase, while others fire late. Further research is needed to unravel the precise molecular mechanisms governing the interplay between the MCM complex and nucleosomes and how these dynamics influence replication timing across different genomic loci. Such insights will deepen our understanding of DNA replication regulation in the context of disease states associated with epigenetic drift, such as aging and cancer.

# Materials and methods

## Yeast strain

Strains used in the study were derived from BY4741 (S288C background, EUROSCARF) and W303 and are provided in *Supplementary file 1*. Cells were grown in standard yeast peptone 2% dextrose media except for HU time course study which employed synthetic media.

Strains containing point mutations in FUN30 were made using a plasmid that contains CAS9 and gRNA targeting the PAM sites adjacent to the desired mutation sites in *FUN30* as described. Oligos containing FUN30-targeting gRNAs were cloned into pML104. pML104 was a gift from John Wyrick (Addgene plasmid # 67638) (*Laughery et al., 2015*). The plasmids, which induce double-strand breaks within FUN30, were co-transformed with DNA blocks containing the repair templates with the desired mutations and altered PAM sites, synthetically synthesized by IDT. *FUN30* DNA was PCR amplified from the resulting transformants and sequenced to verify that the desired mutations have been introduced.

Edu-incorporating yeast strains were derived from W303 and contain four Brdu vectors that express HSV-TK and hENT1 (*Viggiani and Aparicio, 2006*). Plasmids containing BrdU cassette obtained from Addgene (Addgene plasmid # 71789–71792) were integrated at four different auxotrophic loci (*HIS3, TRP1, LEU2,* and *URA3*). Single integration of each Brdu cassette was confirmed via PCR using plasmid-specific primer sets as described (*Viggiani and Aparicio, 2006*), which assured that the strains contain exactly four copies of the BrdU cassette.

## Measurements of rDNA replication timing using S-seq

S-seq experiments were carried out as previously described (*Foss et al., 2017*). Cells were grown to log phase in YEPD media, fixed using 70% (vol/vol) ethanol, subjected to proteinase K digestion, and their DNA stained with Sytox Green as described (*Foss, 2001*). Cells were sorted according to DNA content on a BD Biosciences FACSAria II cell sorter into G1 and S fractions. DNA from a minimum of 1 e-6 cells from each fraction was isolated using the YeaStar Genomic DNA Kit (Zymo Research). DNA was fragmented by sonication and sequenced.

## MNase-seq

We carried out MNase-seq as previously described (*Foss et al., 2019*). Briefly, cells grown to log phase in rich medium, Yeast Peptone Agar with 2% glucose (YEPD), from an overnight 25 mL culture were synchronized with 3 µM alpha factor for 1.5 hr at 30°C. Arrested cells were crosslinked with 1% formaldehyde for 30 min at room temperature water bath with shaking. Formaldehyde was quenched with 125 mM glycine and cells were centrifuged at 3000 rpm for 5 min. Cells were washed twice with water and resuspended in 1.5 mL Buffer Z (1 M sorbitol, 50 mM Tris-HCl pH 7.4) with 1 mM beta-mercaptoethanol (1.1 µL of 14.3 M beta-mercaptoethanol diluted 1:10 in Buffer Z) per 25 mL culture. Cells were treated with 100 µL 20 mg/mL zymolyase at 30°C for 20–30 min. Spheroplasts were centrifuged at 5000 rpm for 10 min and resuspended in 5 mL NP buffer (1 M sorbitol, 50 mM NaCl, 10 mM Tris pH 7.4, 5 mM $MgCl_2$, 1 mM $CaCl_2$) supplemented with 500 µM spermidine, 1 mM beta-mercaptoethanol, and 0.075% NP-40. Nuclei were aliquoted in tubes with varying concentrations of MNase (Worthington), mixed via tube inversion, and incubated at room temperature for 20 min. Chromatin digested with 1.9– 7.5 U MNase per 1/5th of spheroplasts from a 25 mL culture yielded appropriate mono-, di-, tri-nucleosome protected fragments for next-generation sequencing. Digestion was stopped with freshly made 5x stop buffer (5% SDS, 50 mM EDTA) and proteinase K was added (0.2 mg/mL final concentration) for an overnight incubation at 65°C to reverse crosslinking. DNA was extracted with phenol/chloroform and precipitated with ethanol. MNase digestion was analyzed via gel electrophoresis prior to proceeding to library preparation. Sequencing libraries were prepared as described below for ChEC.

## Chromatin endogenous cleavage

ChEC-seq was carried out as previously described (*Foss et al., 2019*; *Foss et al., 2021*). Briefly, MNase was activated by addition of $CaCl_2$ to cells that were permeabilized with digitonin. DNA was extracted from cells using phenol and chloroform, precipitated using salt and ethanol and used to construct sequencing libraries without size fractionation. Cells were centrifuged at 1500×*g* for 2 min, and washed twice in cold Buffer A (15 mM Tris pH 7.5, 80 mM KCl, 0.1 mM EGTA) without

additives. Washed cells were carefully resuspended in 570 µL Buffer A with additives (0.2 mM spermidine, 0.5 mM spermine, 1 mM PMSF, ½ cOmplete ULTRA protease inhibitors tablet, Roche, per 5 mL Buffer A) and permeabilized with 0.1% digitonin in 30°C water bath for 5 min. Permeabilized cells were cooled at room temperature for 1 min and 1/5th of cells were transferred in a tube with freshly made 2x stop buffer (400 mM NaCl, 20 mM EDTA, 4 mM EGTA)/1% SDS solution for undigested control. MNase was activated with 5.5 µL of 200 mM $CaCl_2$ at various times (5 min, and 10 min) and the reaction stopped with 2x stop buffer/1% SDS. Once all time points were collected, proteinase K was added to each of the collected time points and incubated at 55°C water bath for 30 min. DNA was extracted using phenol/chloroform and precipitated with ethanol. MNase digestion was analyzed via gel electrophoresis prior to proceeding to library preparation. Library was prepared as previously described using total DNA, without any fragment size selection (*Foss et al., 2019*; *Foss et al., 2021*).

## Chromatin immunoprecipitation followed by sequencing

Cells were grown to log phase in rich medium, YEPD (1% Yeast Extract, 2% Peptone, 2% glucose), synchronized with 3 µM alpha factor for 1.5 hr at 30°C. G1-arrested cells were released into 200 mM HU YEPD and grown for 1 hr. Cells were crosslinked with 1% formaldehyde for 30 min at room temperature, quenched with 125 mM glycine for 5 min, and centrifuged at 3000 rpm for 5 min. Cell pellets were washed three times with TBS (20 mM Tris-HCl, pH 7.6, 150 mM NaCl). Cell pellets were flash-frozen.

## Chromatin preparation

The cell pellet was thawed in 300 µL Breaking Buffer (100 mM Tris-HCl, pH 8.0, 20% Glycerol) supplemented with protease inhibitors (Pierce Protease inhibitor EDTA-free tablets [Thermo Scientific A32965] and 1 mM PMSF). 300 µL acid-washed glass beads were added to the mixture. Cells were lysed using a BioSpec Mini-Bead-beater in a cold room: 5×30 s pulses with 1 min on ice between rounds. The supernatant was transferred to a new tube and added 600 µL FA buffer (50 mM HEPES-KOH, pH 7.6, 150 mM NaCl, 5 mM EDTA, 1% Triton X-100, 0.1% sodium deoxycholate) with protease inhibitors. The chromatin was sonicated with an F60 Sonic Dismembrator 10 times for 10 s at setting 4, with 1 min rests on ice between rounds. The sonicated chromatin was centrifuged at 14,000 rpm for 15 min at 4°C. The supernatant was transferred and spun again at 14,000 rpm for 15 min at 4°C.

## Immunoprecipitation

For IP, 20 µL of Dynabeads Protein G beads (Invitrogen 10004D) were washed three times with 500 µL PBS-T (0.8% NaCl, 0.144% $Na_2HPO_4$, 0.02% KCl, 0.024% $KH_2PO$, 0.1% Tween 20, pH 7.1), resuspended in PBS-T, and incubated with 5 µL of FLAG M2 (Sigma F1804) antibody for 60 min at room temperature. The antibody-conjugated beads were washed and resuspended in 20 µL FA buffer with protease inhibitors. IP chromatin samples (1 µg) were brought up to 400 µL with FA buffer with protease inhibitors and incubated with antibody-conjugated beads for 90 min at room temperature. Beads were washed three times with FA buffer with protease inhibitors, twice with FA-HS buffer (50 mM HEPES-KOH, pH 7.6, 500 mM NaCl, 5 mM EDTA, 1% Triton X-100, 0.1% sodium deoxycholate with protease inhibitors), and once with RIPA buffer (10 mM Tris-HCl, pH 8.0, 0.25 M LiCl, 0.5% NP-40, 0.5% sodium deoxycholate, 5 mM EDTA with protease inhibitors). Input sample was generated by adding 0.1 µg chromatin in 40 µL FA buffer to 40 µL 2x stop buffer (20 mM Tris-HCl, pH 8.0, 100 mM NaCl, 20 mM EDTA, 1% SDS, 2% Tween 20) and reversed crosslinks by overnight incubation at 65°C.

## Elution and crosslink reversal

Bound chromatin was eluted by incubating the beads in 80 µL 2x stop buffer at 75°C for 10 min. The eluate was collected, and crosslinks were reversed by overnight incubation at 65°C. The IP and input were treated with 4 µL of 20 mg/mL RNase A and incubated at 55°C for 1 hr, followed by 4 µL of 20 mg/mL proteinase K and incubation at 55°C for at least 3 hr. DNA was purified using a MinElute PCR Purification Kit.

## Paired-ends NGS

ChIP samples were prepared for sequencing following the protocol for the Kapa Hyper Kit. For input samples, 2 ng were processed for End Repair. For IP samples, the entire volume was used. PCR amplification was performed with 10–14 cycles based on DNA quantity.

## Quantification of rDNA replication using the thymidine analog EdU

Cells were grown to log phase in rich YEPD medium from a 50 mL culture and synchronized in G1 using 3 µM alpha factor for 2 hr at 30°C. Using 500 µg/mL pronase, arrested cells were released into rich media containing 200 mM HU and 130 µM EdU (Vector Laboratories). Cells were harvested in G1 and after 60 min released into HU. DNA was extracted using YeaStar Genomic DNA Kit (Zymo Research) and sonicated. Separate sequencing primers were ligated to each of the samples. Samples were pooled together, and subjected to click chemistry reaction in 10 mM Tris pH 7.4, 50 mM NaCl, 20 µM biotin-TEG azide (Vector Laboratories), 5 mM THPTA (tris-hydroxypropyltriazolylmethylamine) (Vector Laboratories), 1 mM copper sulfate, and 15 mM sodium ascorbate (total volume 200 µL). Click reaction proceeded for 30 min at room temperature prior to DNA ethanol precipitation. Streptavidin magnetic beads (New England Biolabs) were washed three times with B&W buffer I (100 mM NaCl, 10 mM Tris-HCl pH 8.0, 1 mM EDTA, 0.05% Tween 20, 0.5% SDS) before adding 100 µL of bead suspension to 180 µL DNA plus B&W buffer I. After a 30 min incubation at room temperature while gently inverting tubes, bead bound DNA was washed with 200 µL B&W buffer I followed by 100 µL Stringency wash buffer (0.1% SDS, 0.1% SSC buffer) and 200 µL Wash Buffer II (100 mM NaCl, 10 mM Tris-HCl pH 8.0, 1 mM EDTA, 0.05% Tween 20). DNA was eluted off the beads by denaturation at 98°C for 5 min, followed by cooling at 25°C. Edu-labeled DNA was PCR amplified post-pulldown and subjected to sequencing.

## Hydroxyurea ChEC time course experiment

Cells grown in synthetic media to logarithmic phase were arrested in G1 using alpha factor for 90 min, centrifuged, washed in water to remove alpha factor, and released into media containing 200 mM HU. Cells were harvested in G1 and at indicated time points following release into HU-containing media, chilled at 4°, and analyzed by ChEC as described above.

## qRT-PCR analysis of c-pro transcript and rDNA copy number

RNA was extracted from logarithmically growing cells after spheroplasting using an RNA-Easy column. The sequences of the primers used in qRT-PCR for c-pro and PDA1 mRNA are provided in *Supplementary file 2*. rDNA size was measured by qPCR using DNA that was extracted by phenol-chloroform extraction and ethanol precipitation using the primers listed in As an internal standard, qPCR was done in parallel on the DNA extracted from *fob1* strains with 180 and 35 rDNA copies whose rDNA sizes had been verified by pulsed field gel electrophoresis (*Kwan et al., 2023*).

## ChEC Southern blots for measuring origin licensing

Cells carrying MCM2-MNase were subjected to ChEC as described above, except that the stop SDS was omitted from the stop buffer to allow subsequent spheroplasting. DNA was extracted from spheroplasts using the YeaStar Genomic DNA Kit (Zymo Research), digested with XmnI, separated on 1.75% agarose gels and analyzed by standard Southern blotting technique. The primers used to make the rDNA and *PIK1* control probe are listed in *Supplementary file 2*.

## Acknowledgements

We thank Toshi Tsukiyama, Bonny Brewer, and MK Raghuraman for yeast strains. Research reported in this publication was supported by the National Institute of General Medical Sciences of the National Institutes of Health under Award Number R01GM117446.

## Additional information

### Funding

| Funder | Grant reference number | Author |
|---|---|---|
| National Institutes of Health | R01GM117446 | Antonio Bedalov |

The funders had no role in study design, data collection and interpretation, or the decision to submit the work for publication.

### Author contributions

Carmina Lichauco, Data curation, Formal analysis, Investigation, Methodology, Project administration, Writing – review and editing; Eric J Foss, Conceptualization, Formal analysis, Investigation, Visualization, Writing – original draft, Writing – review and editing; Tonibelle Gatbonton-Schwager, Data curation, Methodology; Nelson F Athow, Data curation, Investigation; Brandon Lofts, Data curation, Investigation, Methodology; Robin Acob, James J Marquez, Investigation; Erin Taylor, Investigation, Methodology; Uyen Lao, Data curation, Investigation, Project administration; Shawna Miles, Investigation, Project administration; Antonio Bedalov, Conceptualization, Supervision, Funding acquisition, Writing – original draft, Project administration, Writing – review and editing

### Author ORCIDs

Eric J Foss (i) https://orcid.org/0000-0002-5553-9412
Antonio Bedalov (i) https://orcid.org/0000-0002-7373-8255

Reviewer #1 (Public review): https://doi.org/10.7554/eLife.97438.4.sa1
Reviewer #2 (Public review): https://doi.org/10.7554/eLife.97438.4.sa2
Reviewer #3 (Public review): https://doi.org/10.7554/eLife.97438.4.sa3
Author response https://doi.org/10.7554/eLife.97438.4.sa4

## Additional files

### Supplementary files

MDAR checklist

Supplementary file 1. Table with the description of the yeast strains.

Supplementary file 2. Table with primer sequences.

### Data availability

Sequencing data have been deposited in GEO under accession code GSE285768. All the data analyzed in this study are included in the manuscript and the supporting files.

The following dataset was generated:

| Author(s) | Year | Dataset title | Dataset URL | Database and Identifier |
|---|---|---|---|---|
| Bedalov A, Foss EJ | 2025 | Sir2 and Fun30 regulate ribosomal DNA replication timing via MCM helicase positioning and nucleosome occupancy | http://www.ncbi.nlm.nih.gov/geo/query/acc.cgi?acc=GSE285768 | NCBI Gene Expression Omnibus, GSE285768 |

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
