## [Editor Report · eLife Assessment]

This **valuable** study is a detailed investigation of how chromatin structure influences replication origin function in yeast ribosomal DNA, with a focus on the role of the histone deacetylase Sir2 and the chromatin remodeler Fun30. The paper shows that Sir2 does not affect origin licensing but rather affects local transcription and nucleosome positioning which correlates with increased origin firing. Overall, the evidence is **convincing** and the model is plausible.

---

## [Referee Report · Reviewer #1 (Public review)]

This paper presents a mechanistic study of rDNA origin regulation in yeast by SIR2. Each of the ~180 tandemly repeated rDNA gene copies contains a potential replication origin. Early-efficient initiation of these origins is suppressed by Sir2, reducing competition with origins distributed throughout the genome for rate-limiting initiation factors. Previous studies by these authors showed that SIR2 deletion advances replication timing of rDNA origins by a complex mechanism of transcriptional de-repression of a local PolII promoter causing licensed origin proteins (MCMcomplexes) to re-localize (slide along the DNA) to a different (and altered) chromatin environment. In this study, they identify a chromatin remodeler, FUN30, that suppresses the sir2∆ effect, and remarkably, results in a contraction of the rDNA to about one-quarter it's normal length/number of repeats, implicating replication defects of the rDNA. Through examination of replication timing, MCM occupancy and nucleosome occupancy on the chromatin in sir2, fun30, and double mutants, they propose a model where nucleosome position relative to the licensed origin (MCM complexes) intrinsically determines origin timing/efficiency. While their interpretations of the data are largely reasonable and can be interpreted to support their model, a key weakness is the connection between Mcm ChEC signal disappearance and origin firing. While the cyclical chromatin association-dissociation of MCM proteins with potential origin sequences may be generally interpreted as licensing followed by firing, dissociation may also result from passive replication and as shown here, displacement by transcription and/or chromatin remodeling. Moreover, linking its disappearance from chromatin in the ChEC method with such precise resolution needs to be validated against an independent method to determine the initiation site(s). Differences in rDNA copy number and relative transcription levels also are not directly accounted for, obscuring a clearer interpretation of the results. Nevertheless, this paper makes a valuable advance with the finding of Fun30 involvement, which substantially reduces rDNA repeat number in sir2∆ background. The model they develop is compelling and I am inclined to agree, but I think the evidence on this specific point is purely correlative and a better method is needed to address the initiation site question. The authors deserve credit for their efforts to elucidate our obscure understanding of the intricacies of chromatin regulation.

Overall, the paper is improved by providing additional data and improved analysis. The paper nicely characterizes the effect of Fun30. The model is reasonable but remains lacking in precise details of mechanism.

---

## [Referee Report · Reviewer #2 (Public review)]

Summary:

In this manuscript, the authors follow up on their previous work showing that in the absence of the Sir2 deacetylase the MCM replicative helicase at the rDNA spacer region is repositioned to a region of low nucleosome occupancy. Here they show that the repositioned displaced MCMs have increased firing propensity relative to non-displaced MCMs. In addition, they show that activation of the repositioned MCMs and low nucleosome occupancy in the adjacent region depend on the chromatin remodeling activity of Fun30.

Strengths:

The paper provides new information on the role of a conserved chromatin remodeling protein in regulation of origin firing and in addition provides evidence that not all loaded MCMs fire and that origin firing is regulated at a step downstream of MCM loading.

Comments on revisions:

The authors have addressed my concerns with the addition of new experiments and analysis.

---

## [Referee Report · Reviewer #3 (Public review)]

Summary:

Heterochromatin is characterized by low transcription activity and late replication timing, both dependent on the NAD-dependent protein deacetylase Sir2, the founding member of the sirtuins. This manuscript addresses the mechanism by which Sir2 delays replication timing at the rDNA in budding yeast. Previous work from the same laboratory (Foss et al. PLoS Genetics 15, e1008138) showed that Sir2 represses transcription-dependent displacement of the Mcm helicase in the rDNA. In this manuscript, the authors show convincingly that the repositioned Mcms fire earlier and that this early firing partly depends on the ATPase activity of the nucleosome remodeler Fun30. Using read-depth analysis of sorted G1/S cells, fun30 was the only chromatin remodeler mutant that somewhat delayed replication timing in sir2 mutants, while nhp10, chd1, isw1, htl1, swr1, isw2, and irc5 had no effect. The conclusion was corroborated with orthogonal assays including two-dimensional gel electrophoresis and analysis of EdU incorporation at early origins. Using an insightful analysis with an Mcm-MNase fusion (Mcm-ChEC), the authors show that the repositioned Mcms in sir2 mutants fire earlier than the Mcm at the normal position in wild type. This early firing at the repositioned Mcms is partially suppressed by Fun30. In addition, the authors show Fun30 affects nucleosome occupancy at the sites of the repositioned Mcm, providing a plausible mechanism for the effect of Fun30 on Mcm firing at that position. However, the results from the MNAse-seq and ChEC-seq assays are not fully congruent for the fun30 single mutant. Overall, the results support the conclusions providing a much better mechanistic understanding how Sir2 affects replication timing at rDNA,

Strengths:

(1) The data clearly show that the repositioned Mcm helicase fires earlier than the Mcm in the wild type position.

(2) The study identifies a specific role for Fun30 in replication timing and an effect on nucleosome occupancy around the newly positioned Mcm helicase in sir2 cells.

Comments on revisions:

In the previous revision the authors addressed my concerns and improved the manuscript and the presentation of the data. All my recommendations were implemented.

---

## [Author Response]

The following is the authors’ response to the previous reviews.

**eLife Assessment**
This valuable study is a detailed investigation of how chromatin structure influences replication origin function in yeast ribosomal DNA, with a focus on the role of the histone deacetylase Sir2 and the chromatin remodeler Fun30. Convincing evidence shows that Sir2 does not affect origin licensing but rather affects local transcription and nucleosome positioning which correlates with increased origin firing. Overall, the evidence is solid and the model plausible. However, the methods employed do not rigorously establish a key aspect of the mechanism where initiation precisely occurs or rigorously exclude alternative models and the effect of Sir2 on transcription is not re-examined in the fun30 context.

Clarification on Sir2 Effect on Transcription in the *fun30* Context

We appreciate the reviewers’ thorough assessment but would like to clarify that the effect of Sir2 on transcription in the *fun30* context was addressed in both the original and revised manuscripts. However, we recognize that the presentation of the qPCR results may have been unclear, as we initially plotted absolute transcript levels without normalizing for rDNA array size differences among the genotypes. We have now corrected this.

After normalizing for copy number variations, the qPCR data show that the *sir2 fun30* double mutant results in a ~40-fold increase in C-pro transcription relative to WT, compared to a 4-fold and 19-fold increase in fun30 and sir2 single mutants, respectively (Figure 5, figure supplement 6). These results have been discussed in the manuscript result section, where we note that "C-pro RNA levels were approximately twice as high in *sir2 fun30* compared to *sir2* cells when adjusted for rDNA size differences." This observation is critical for addressing both alternative models of MCM disappearance and for pinpointing transcription initiation sites, as detailed in the following sections.

**Public Reviews:**

**Reviewer #1 (Public review):**
Summary:This paper presents a mechanistic study of rDNA origin regulation in yeast by SIR2. Each of the ~180 tandemly repeated rDNA gene copies contains a potential replication origin. Earlyefficient initiation of these origins is suppressed by Sir2, reducing competition with origins distributed throughout the genome for rate-limiting initiation factors. Previous studies by these authors showed that SIR2 deletion advances replication timing of rDNA origins by a complex mechanism of transcriptional de-repression of a local PolII promoter causing licensed origin proteins (MCMcomplexes) to re-localize (slide along the DNA) to a different (and altered) chromatin environment. In this study, they identify a chromatin remodeler, FUN30, that suppresses the sir2∆ effect, and remarkably, results in a contraction of the rDNA to about onequarter it's normal length/number of repeats, implicating replication defects of the rDNA. Through examination of replication timing, MCM occupancy and nucleosome occupancy on the chromatin in sir2, fun30, and double mutants, they propose a model where nucleosome position relative to the licensed origin (MCM complexes) intrinsically determines origin timing/efficiency. While their interpretations of the data are largely reasonable and can be interpreted to support their model, a key weakness is the connection between Mcm ChEC signal disappearance and origin firing. While the cyclical chromatin association-dissociation of MCM proteins with potential origin sequences may be generally interpreted as licensing followed by firing, dissociation may also result from passive replication and as shown here, displacement by transcription and/or chromatin remodeling. Moreover, linking its disappearance from chromatin in the ChEC method with such precise resolution needs to be validated against an independent method to determine the initiation site(s). Differences in rDNA copy number and relative transcription levels also are not directly accounted for, obscuring a clearer interpretation of the results. Nevertheless, this paper makes a valuable advance with the finding of Fun30 involvement, which substantially reduces rDNA repeat number in sir2∆ background. The model they develop is compelling and I am inclined to agree, but I think the evidence on this specific point is purely correlative and a better method is needed to address the initiation site question. The authors deserve credit for their efforts to elucidate our obscure understanding of the intricacies of chromatin regulation. At a minimum, I suggest their conclusions on these points of concern should be softened and caveats discussed. Statistical analysis is lacking for some claims.Strengths are the identification of FUN30 as suppressor, examination of specific mutants of FUN30 to distinguish likely functional involvement. Use of multiple methods to analyze replication and protein occupancies on chromatin. Development of a coherent model.Weaknesses are failure to address copy number as a variable; insufficient validation of ChEC method relationship to exact initiation locus; lack of statistical analysis in some cases.Review of revised version and response letter:In the response, the authors make some improvements by better quantifying 2D gels, adding some missing statistical analyses, analyzing the effect of fun30 on rDNA replication in strains with reduced rDNA copy number, and using ChIP-seq of MCMs to support the ChEC-seq data. However, these additions do not address the main issue that is at the heart of their model: where initiation precisely occurs and whether the location is altered in the mutant(s). Thus, mechanistic insight is limited.

We discuss the issue regarding the initiation site below.

Under the section "Addressing Alternative Explanations", the authors claim that processes like transcription and passive replication cannot affect the displaced complex specifically. Why? They are not on same DNA (as mentioned in the Fig 1 legend).

Premature origin activation, not transcription, drives the disappearance of repositioned MCM complexes in *sir2* mutants in HU.

Indeed, the reviewer is correct in suggesting that C-pro transcription confined to rDNA units with repositioned MCM complexes could selectively displace those complexes, potentially explaining the selective disappearance of displaced MCMs in *sir2* cells. However, our analysis of C-pro transcription and MCM occupancy in G1 versus HU across the genotypes allows us to rule out this possibility.

We show that the fraction of repositioned MCMs in G1 cells is proportional to the level of C-pro transcription (WT < *fun30* << *sir2* < *sir2 fun30*), consistent with the involvement of transcription in the repositioning process during MCM loading in G1. Accordingly, with approximately twice the transcription in *sir2 fun30* compared to *sir2*, we observe more repositioned MCMs in *sir2 fun30* cells than in *sir2* cells in G1 (Fig 5C).

However, if the disappearance of repositioned MCMs in HU were solely due to C-pro transcription rather than origin activation, we would expect the repositioned MCMs to disappear more quickly in *sir2 fun30* cells. Contrary to this expectation, our data show that repositioned MCM complexes are more stable in *sir2 fun30* mutants compared to *sir2* mutants, indicating that transcription is not the primary factor in the disappearance of displaced MCM complexes in HU; rather, rDNA origin activation appears to be the key factor.

Replication initiation site in *sir2*. Using multiple independent approaches, including 2D gels, ChIP-seq, and EdU incorporation, we have demonstrated that *rDNA* origins fire prematurely in *sir2* mutants, a conclusion that the reviewer does not contest. Once an origin fires, the MCM signal disappears from the site of its initial deposition, as expected, and this is confirmed in our MCM ChIP and HU ChEC data, both at rDNA origins and across the genome.

Given that the majority of MCM complexes in *sir2* mutants are *repositioned***,** it is expected that these repositioned complexes disappear following premature origin activation. With less than half of the licensed origins (or <30% of total rDNA copies) retaining MCM at non-repositioned sites in *sir2* mutants, if only these non-repositioned complexes were firing, and the repositioned MCM complexes were disappearing via mechanisms other than replication initiation (e.g., transcription), rDNA replication in *sir2* mutants would be severely compromised rather than accelerated. Given this, and the strong experimental evidence that repositioned MCM complexes fire prematurely, continued focus on alternative explanations for MCM complex disappearance seems unwarranted.

We present this analysis in the results section as follows:

“Finally, although deletion of *FUN30* could suppress replication initiation at the rDNA either by inhibiting the firing of the active, repositioned MCM complex or by preventing MCM repositioning to the "active location" in the first place, our results suggest that suppression occurs through the former mechanism. Consistent with previous reports that *fun30* mutants are deficient in transcriptional silencing (Neves-Costa et al. 2009), C-pro RNA levels were approximately twice as high in *sir2 fun30* cells compared to *sir2* cells when adjusted for rDNA size (Figure 5—figure supplement 6).

Moreover, deletion of *FUN30* shifts the distribution toward the repositioned MCM location over the non-repositioned one in G1 cells (Figure 5C), aligning with the increased C-pro transcription observed in *fun30* mutants. This shift is evident in both *sir2* and *SIR2* cells. Despite the increased transcription-mediated repositioning in *sir2 fun30* cells compared to *sir2* cells during G1, repositioned MCM persists longer in *sir2 fun30* cells than in *sir2* cells after release into HU. Additionally, *sir2 fun30* mutants exhibit reduced MCM accumulation at the RFB compared to *sir2* mutants after release into HU, supporting the conclusion that MCM disappearance in HU reflects origin activation rather than transcription-mediated displacement.”

The model in Fig 7 implies that initiation sites are different in WT versus the mutants and this determines their timing/efficiency. But they also suggest that the same site might be used with different efficiencies in this response. I agree that both are possibilities and are not resolved.

Adjustment of the model to account for repositioned MCMs in WT cells

In Figure 5—figure supplement 5, we demonstrate that even in WT cells, a small fraction of repositioned MCMs (~5%) can be detected, and that these repositioned MCM complexes disappear prematurely. However, because this represents a very small fraction of MCMs in WT cells, we initially did not include it in our overall model in Figure 7. In light of the reviewer's comment, we have now revised the model to incorporate this detail.

Supporting their model requires better resolution to determine the actual replication initiation site. While this may be challenging, it should be feasible with methods to map nascent strands like DNAscent, or Okazaki fragment mapping.

The initiation site in *sir2* mutants has been thoroughly analyzed and supported by extensive experimental data, as discussed above. While high-resolution techniques such as DNAscent or Okazaki fragment mapping could potentially offer another layer of validation, the likelihood of obtaining finer detail that would change the conclusions is minimal. The methods we employed provide sufficient resolution to pinpoint the initiation site, and our results align consistently with established replication models.

Further experimentation would not only be redundant but also unlikely to provide new insights beyond revalidation. Given the strength of our current data, we believe the conclusions regarding replication initiation are robust and well-supported, making additional experiments unnecessary at this stage. Our priority is to focus on advancing other aspects of the research that require deeper exploration.

The 2D gel analysis of strains with reduced rDNA copy numbers adequately addresses the copy number variable with regard to the replication effect.Overall, the paper is improved by providing additional data and improved analysis. The paper nicely characterizes the effect of Fun30. The model is reasonable but remains lacking in precise details of mechanism.
**Reviewer #2 (Public review):**
Summary:In this manuscript, the authors follow up on their previous work showing that in the absence of the Sir2 deacetylase the MCM replicative helicase at the rDNA spacer region is repositioned to a region of low nucleosome occupancy. Here they show that the repositioned displaced MCMs have increased firing propensity relative to non-displaced MCMs. In addition, they show that activation of the repositioned MCMs and low nucleosome occupancy in the adjacent region depend on the chromatin remodeling activity of Fun30.Strengths:The paper provides new information on the role of a conserved chromatin remodeling protein in regulation of origin firing and in addition provides evidence that not all loaded MCMs fire and that origin firing is regulated at a step downstream of MCM loading.Weaknesses:The relationship between the authors results and prior work on the role of Sir2 (and Fob1) in regulation of rDNA recombination and copy number maintenance is not explored, making it difficult to place the results in a broader context. Sir2 has previously been shown to be recruited by Fob1, which is also required for DSB formation and recombination-mediated changes in rDNA copy number. Are the changes that the authors observe specifically in fun30 sir2 cells related to this pathway? Is Fob1 required for the reduced rDNA copy number in fun30 sir2 double mutant cells?
**Reviewer #3 (Public review):**
Summary:Heterochromatin is characterized by low transcription activity and late replication timing, both dependent on the NAD-dependent protein deacetylase Sir2, the founding member of the sirtuins. This manuscript addresses the mechanism by which Sir2 delays replication timing at the rDNA in budding yeast. Previous work from the same laboratory (Foss et al. PLoS Genetics 15, e1008138) showed that Sir2 represses transcription-dependent displacement of the Mcm helicase in the rDNA. In this manuscript, the authors show convincingly that the repositioned Mcms fire earlier and that this early firing partly depends on the ATPase activity of the nucleosome remodeler Fun30. Using read-depth analysis of sorted G1/S cells, fun30 was the only chromatin remodeler mutant that somewhat delayed replication timing in sir2 mutants, while nhp10, chd1, isw1, htl1, swr1, isw2, and irc5 had no effect. The conclusion was corroborated with orthogonal assays including two-dimensional gel electrophoresis and analysis of EdU incorporation at early origins. Using an insightful analysis with an Mcm-MNase fusion (Mcm-ChEC), the authorsshow that the repositioned Mcms in sir2 mutants fire earlier than the Mcm at the normal position in wild type. This early firing at the repositioned Mcms is partially suppressed by Fun30. In addition, the authors show Fun30 affects nucleosome occupancy at the sites of the repositioned Mcm, providing a plausible mechanism for the effect of Fun30 on Mcm firing at that position. However, the results from the MNAse-seq and ChEC-seq assays are not fully congruent for the fun30 single mutant. Overall, the results support the conclusions providing a much better mechanistic understanding how Sir2 affects replication timing at rDNA,Strengths(1) The data clearly show that the repositioned Mcm helicase fires earlier than the Mcm in the wild type position.(2) The study identifies a specific role for Fun30 in replication timing and an effect on nucleosome occupancy around the newly positioned Mcm helicase in sir2 cells.Weaknesses(1) It is unclear which strains were used in each experiment.(2) The relevance of the fun30 phospho-site mutant (S20AS28A) is unclear.(3) For some experiments (Figs. 3, 4, 6) it is unclear whether the data are reproducible and the differences significant. Information about the number of independent experiments and quantitation is lacking. This affects the interpretation, as fun30 seems to affect the +3 nucleosome much more than let on in the description.
**Recommendations for the authors:**
**Reviewer #2 (Recommendations for the authors)**:The authors have addressed my concerns by the addition of new experiments and analysis.One point remains unclear regarding additional support for the Mcm-ChEC results using ChIP experiments to verify whether MCM redistributes in sir2D cells. In their rebuttal, the authors state that, "New supporting based evidence: ChIP at rDNA Origins. Our ChIP analysis also shows that the disappearance of the MCM signal at rDNA origins in sir2Δ cells released into HU is accompanied by signal accumulation at the replication fork barrier (RFB), indicative of stalled replication forks at this location (Figure 5 figure supplement 3)...." The ChIP data in Figure 5 supplement 3 show accumulation of the Mcm2 ChIP signal to the left of the RFB in sir2D cells but it doesn't look like there is any decrease in the MCM signal in sir2D relative to wild-type cells for the peak C-Pro. There is a new MCM peak suggesting perhaps a new MCM loading event.

Figure 5 figure supplement 3 shows the relative abundance of the MCM ChIP signal across the ~2 kb rDNA region, spanning from the MCM loading site at the rDNA origin (on the left) to the replication fork barrier (RFB) on the right. The MCM-ChIP data are normalized to the highest signal within this rDNA region rather than across the entire genome, meaning that only the relative abundance of MCM within this region is represented, and not comparisons between different conditions. We have now presented the results with the same axes for both alpha factor and HU.

In wild-type (WT) cells, the MCM signal remains primarily at the initial loading site. However, in *sir2* mutants, a significant portion of the MCM signal shifts rightward, consistent with rDNA origin activation and the movement of MCM along with the progressing replication fork. While some replication forks stall at the RFB, others are positioned between the MCM loading site and the RFB. The additional MCM peak observed does not represent a new MCM loading event, as the experiment was conducted during S-phase, when new MCM loading is not possible.

**Reviewer #3 (Recommendations for the authors):**
In this revision the authors addressed my concerns and improved the manuscript and the presentation of the data. All my recommendations were implemented.